

# Air quality in the eastern United States and Eastern Canada for 1990-2015: 25 years of change in response to emission reductions of SO₂ and NOₓ in the region

Jian Feng[1*], Elton Chan[2] and Robert Vet[1]

[1]Air Quality Measurement and Analysis Research Section, Atmospheric Science and Technology Directorate, Environment and Climate Change Canada, Toronto, Canada

[2]Measurements, Modelling and Interpretation Section, Atmospheric Science and Technology Directorate, Environment and Climate Change Canada, Toronto, Canada

*Correspondence to: jian.feng@canada.ca

## Abstract

SO₂ and NOₓ are precursors to form sulfate, nitrate and ammonium particles, which account for more than 50% of PM2.5 mass in the eastern US and Eastern Canada, and are dominant components of
PM2.5 during many smog events. $H_2SO_4$ and $HNO_3$, formed from oxidation of SO₂ and NOₓ respectively, are the main sources of acid deposition through wet and dry depositions. NOₓ is also a precursor to the formation of tropospheric O₃, which is an important atmospheric oxidant and is also essential for the formation of other atmospheric oxidants, such as OH and $H_2O_2$.

In the past 26 years from 1990 to 2015, emissions of SO₂ and NOₓ in US were significantly reduced
from 23.1 and 25.2 million tons/year in 1990   to 3.7 and 11.5 million tons/year in 2015 respectively. In Canada, SO₂ and NOₓ were reduced by 63% and 33% from 1990 to 2014. In response to the significant reduction of SO₂ and NOₓ emissions, air quality in the eastern US and Eastern Canada improved tremendously during 1990-2015. In this study, we analyzed surface air concentrations of $SO_4^{2-}$, $NO_3^-$, $NH_4^+$, $HNO_3$ and SO₂ measured weekly by the Clean Air Status and Trends Network (CASTNET) in the US
and measured daily from the Canadian Air and Precipitation Monitoring Network (CAPMoN) in Canada to reveal the temporal and spatial changes of each species during the 25-year period. For the whole



the eastern US and Eastern Canada, the annual mean concentrations of $SO_4^{2-}$, $NO_3^-$, $NH_4^+$, $HNO_3$, $SO_2$ and $TNO_3$ ($NO_3^-$ + $HNO_3$, expressed as the mass of equivalent $NO_3^-$) were reduced by 73.3%, 29.1%, 67.4%, 65.8%, 87.6% and 52.6% respectively from 1990 to 2015. In terms of percentage, reduction of all species except $NO_3^-$ was spatially uniform; reduction of $SO_2$ and $HNO_3$ was similar in warm season

(May-October) and cold season (November-April), and reduction of $SO_4^{2-}$, $NO_3^-$ and $NH_4^+$ was more significant in warm season than in cold season. Reduction of $SO_4^{2-}$ and $SO_2$ mainly occurred in 1989-1995 and 2007-2015 during warm season, and in 1989-1995 and 2005-2015 during cold season. Reduction of $NO_3^-$ mainly occurred in the Midwest after 2000. Other than in the Midwest, $NO_3^-$ had very little change during cold season for the period. The reduction of $NH_4^+$ generally followed the

reduction trend of $SO_4^{2-}$, especially after 2000 the temporal trend of $NH_4^+$ was almost identical to that of $SO_4^{2-}$. The ratio of $S$ in $SO_4^{2-}$ to total $S$ in $SO_4^{2-}$ and $SO_2$, as well as the ratio of $NO_3^-$ to $TNO_3$ increased by more than 50% during the period. This indicates that much more percentage of $SO_2$ was oxidized to $SO_4^{2-}$, and much more percentage of $HNO_3$ was neutralized to $NH_4NO_3$ in the region near the end of the period.

## 1. Introduction

Gases and particulate matter released into the air through anthropogenic activities can pollute the air and deteriorate the air quality locally, regionally, and continentally. Air pollution, which can decrease lung function, causing the development of asthma, bronchitis and lung cancer

(http://www.who.int/mediacentre/factsheets/fs313/en/), is considered as a major environmental risk to human health by the World Health Organization (WHO). Air pollution is also linked to stroke and heart disease and improvement of air quality can significantly reduce the PM2.5- and $O_3$-related mortality burden (Zhang et al., 2018). When emitted gases and particulate matter or secondary pollutants formed in the air from emissions are brought to the Earth's surface through dry and/or wet

deposition, they pose a risk to the established ecosystem through acid rain as well as excessive deposition of nitrogen and sulfur. Air pollution also affects long term climate through scattering and absorption of solar radiation by directly emitted or secondarily formed aerosols in the air (Haywood





and Shine, 1995; Yu et al., 2006). In some heavily polluted regions, even local weather can be affected due to the change of energy budgets in the atmosphere and at the Earth's surface (Kajino et al., 2017).

In order to control air pollution, the US passed the Clean Air Act (CAA) of 1963. Major amendments to

the law were passed in 1970, 1977 and 1990. The Amendments to CAA of 1990 addressed acid deposition, ozone depletion, and toxic air pollution. Specifically Title IV of 1990 Amendments to CAA, also known as acid deposition control, targeted emission reduction of two acid deposition precursors, $SO_2$ and $NO_X$, which along with CO, $O_3$, Pb and particulate matter, are among the 6 species designated as criteria pollutants by United States Environmental Protection Agency (US EPA). $SO_2$ and $NO_x$ in the

air can be oxidized to form acid $H_2SO_4$ and $HNO_3$, which in turn can react with $NH_3$ to form fine particulate matter (PM2.5), and with crustal material or sea salts to form coarse particles (Yoshizumi and Hoshi, 1985; Zhuang et al., 1999). $NO_X$, together with volatile organic compounds (VOCs), also participate in formation of tropospheric $O_3$, which is another criteria pollutant and an important atmospheric oxidant. Title IV of Clear Air Act 1990 specifically targets $SO_2$ and $NO_x$ emissions from

stationary fuel combustion facilities. The first phase of the Title IV of 1990 CAA Amendment, which was implemented on January 1, 1995, requires 110 power plants to reduce the $SO_2$ emissions to a level calculated as the product of an emissions rate of 2.5 lbs of $SO_2$/mmBtu times an average of their 1985-1987 fuel use. The second phase, which took effect on January 1, 2000, requires approximately 2000 utilities to reduce $SO_2$ emission to a level of 1.2 lbs of $SO_2$/mmBtu times the average of their 1985-

1997 fuel use. Since 1990, national emission of $SO_2$ in US decreased steadily from 23.1 million tons in 1990 to 21.3 million tons in 1994, and dropped significantly to 18.6 million tons in 1995 due to the first phase implementation of Title IV of 1990 CAA Amendments. The $SO_2$ emissions underwent small increase during 1996-1998 to 18.9 million tons in 1998, and then continued the steady decrease to 14.5 million tons in 2005. From 2005 to 2012, the decrease of the emissions was accelerated with an

annual reduction rate of 1.34 million tons /year during the period. The emission of $SO_2$ was leveled off during 2012-2015. In 1990, 87.9% of $SO_2$ emission was from stationary fuel combustion facilities, 2%



from on-road vehicles, and 2% from off-road mobile. By 2007, $SO_2$ emission from on-road vehicles was totally eliminated due to cleaner gasoline. In 2014, of the 4.9 million tons of total $SO_2$ emissions, stationary fuel combustion, off-road mobile, and industrial and other processes contributed 4.1, 0.1, and 0.7 million tons respectively (https://gispub.epa.gov/air/trendsreport/2016/).

$NO_x$ forms in air when nitrogen reacts with oxygen under high temperature. Anthropogenic emission of $NO_x$ is mainly due to stationary fuel combustion, on-road vehicles and off-road mobile operations. Nationwide in the US, they contributed 10.9, 9.6 and 3.8 million tons of the total 25.2 million tons of $NO_x$ in 1990. Changes in $NO_x$ emission during 1990s were relatively small (Butler et al., 2003). Total

10 $NO_x$ emission remained pretty constant from 1990 to 1998. From 1999 there was a decreasing of $NO_x$ emission from stationary fuel combustion, due to the implementation of Title IV of 1990 CAA Amendment, which also stipulated the reduction of $NO_x$ emission from power plants, and took effect in 1996, as well as the implementation of the $NO_x$ Budget Trading Program (NBP) that started in 2003 and was created to reduce $NO_x$ emissions from power plants and other large combustion sources in

the eastern US during warm months (https://www.epa.gov/airmarkets/nox-budget-trading-program). The NBP was replaced by the ozone season $NO_x$ program under the Clean Air Interstate Rule in 2009. The $NO_x$ emissions from stationary combustion facilities decreased steadily from 10.4 million tons in 1998 to 3.6 million tons in 2012, then remained relatively unchanged thereafter (https://gispub.epa.gov/air/trendsreport/2016/). Emissions of $NO_x$ from on-road vehicles declined

slowly from 1990 until 2001, and there was a sharp increase in 2022. After 2022, on-road emission of $NO_x$ decreased continuously and steadily. The trend of $NO_x$ emission from off-road mobile was generally increased during the period from 1990-2002, up from 3.8 tons to 4.9 tons, but after that it was reduced gradually to 2.7 tons in 2014. Combining the emissions from stationary fuel combustion, on-road vehicles and off-road mobiles, the nationwide emissions of $NO_x$ in the US changed little during

1990-1998, decreased during 1998-2001, increased in 2002, and then decreased steeply up to recent years.



Air quality trends during the past few decades, especially since 1990, have been reported for Europe and East Asia (Colette et al., 2011; Guerreiro et al., 2014; Colette et al., 2015; Colette et al., 2017; Wang et al., 2013). For the eastern part of US and Canada, trends of air quality after 1990 have been

reported in previous studies for $O_3$ (Chan and Vet, 2009), $O_3$ and nitrate (Butler et al., 2011), particulate $SO_4^{2-}$ (Hand et al., 2012) and air quality and atmospheric deposition (Sickles and Shadwick, 2007; Sickles and Shadwick, 2015, Cheng and Zhang, 2017). Sickles and Shadwick (2007, 2015) compared the 5-year averages of air quality and atmospheric deposition in the eastern US for 1990-2004 and 1990-2009. Cheng and Zhang (2017) reported the temporal trends of annual concentration

of air pollutants from 31 Canadian rural locations, most of which were located in Eastern Canada. Aas et al. (2019) reported global and regional trends of atmospheric sulfur for 1990-2015, and found North America and East Asia had the largest reductions of sulfur emissions during the late part of the period. In this study, we analyze the surface air concentration data measured weekly by the Clean Air Status and Trends Network (CASTNET) in the US and measured daily from the Canadian Air and Precipitation

Monitoring Network (CAPMoN) in Canada to reveal the detailed temporal and spatial trends of air quality from 1989-2016. These trends are not only important for the assessment of the improvement of air quality due to emissions reductions, but also are essential for the evaluations of chemical transportation models. The analysis will answer the questions of: (1) what are the trends of air pollutants over the eastern US and Eastern Canada following the significant reductions of $SO_2$ and $NO_x$

emissions during 1990-2015; (2) the physical and chemical mechanisms responsible for the trends. We will look at the air concentration of gases $SO_2$ and $HNO_3$, and particulates $SO_4^{2-}$, $NO_3^-$, and $NH_4^+$, which are either due to direct emissions of $SO_2$, or due to the oxidation of $SO_2$ and $NO_x$ as well as reaction of these oxidants with $NH_3$. Specifically, we will look at the followings for species of $SO_4^{2-}$, $SO_2$, $NH_4^+$, $NO_3^-$, $HNO_3$ and $TNO_3$ ( $NO_3^-$ + $HNO_3$, expressed as equivalent $NO_3^-$) :

(1)  temporal and spatial trends in the eastern US and Eastern Canada;

         (2)  10-year and 25-year changes for the periods of 1990-2000 and 1990-2015;



(3) differences in trends in cold and warm seasons;

(4) time series of the annual means during warm and cold seasons;

(5) long-term trends derived from polynomial regressions.

We will also look at correlations between $SO_4^{2-}$ and $SO_2$, the ratio of sulfur ($RSO_4$) in $SO_4^{2-}$ to total sulfur in $SO_4^{2-}$ and $SO_2$ in the air, the ratio of nitrogen ($RNO_3$) in $NO_3^-$ to $TNO_3$, and their changes during the period.

## 2. Networks of measurement and clustering of measurement sites in the eastern US
and Eastern Canada

### 2.1 CASTNET and CAPMoN

Monitoring background-level ambient pollutants is essential for assessing regional air quality. In the US
and Canada, this long-term mandate of monitoring air quality in rural areas is fulfilled by the two monitoring networks, the Clean Air Status and Trends Network (CASTNET), and the Canadian Air and Precipitation Monitoring Network (CAPMoN) respectively.

CASTNET is a monitoring network managed and operated by the U.S. Environment Protection Agency
(EPA) in cooperation with some other federal, state and local partners. The network was established under the 1990 Clean Air Act Amendments to assess the trends of acidic deposition due to emission reduction programs. The network makes weekly measurements of gases ($SO_2$ and $HNO_3$) and particulates ($SO_4^{2-}$, $NO_3^-$, $NH_4^+$, $Mg^{2+}$, $Ca^{2+}$, $Na^+$, and $Cl^-$), as well as hourly measurements of $O_3$. At selected sites, it also measures hourly concentration of NO, reactive nitrogen ($NO_y$), $SO_2$, and CO.



CAPMoN is a monitoring network operated by Environment and Climate Change of Canada. The network began operation in 1983, but one of the two monitoring networks integrated into CAPMoN, Air Precipitation Network (APN), can go as far back as 1978. The network measures both daily air concentration of pollutants through filter sampling, and daily wet deposition through collection of

5   precipitation samples at the ground level. The daily air concentration measurement by CAPMoN also include gases ($SO_2$ and $HNO_3$) and particulates ($SO_4^{2-}$, $NO_3^-$, $NH_4^+$, $Mg^{2+}$, $Ca^{2+}$, $Na^+$, and $Cl^-$), similar to CASTNET's weekly measurement. CAPMoN also measures hourly air concentration of $O_3$, $NO_y$, and gaseous Hg at selected sites. More details about the CAPMoN dataset can be found in Cheng and Zhang (2017).

**2.2  Region-clustering of CASTNET and CAPMoN sites in the eastern US and Eastern Canada**

In the eastern US (EUS) and Eastern Canada (EC) there are significant spatial differences in emissions of $SO_2$, $NO_x$ and $NH_3$. This results in distinctive regional patterns of air concentration of $SO_4^{2-}$, $NO_3^-$,

$NH_4^+$, $HNO_3$ and $SO_2$. In this study, we used the cold season (November to April) mean concentrations of $NO_3^-$ and $SO_2$, supplementing with the ratio of $RNO_3$, as the criteria to cluster CASTNET and CAPMoN sites into 4 different regions. The reasons for selecting cold season are that: (1) $NO_3^-$ is mainly in form of $NH_4NO_3$ (Zhang et al., 2008), and it is much more thermodynamically stable than in warm season; (2) $SO_2$ is much less being oxidized in cold season than in warm season, therefore the air concentration

of $SO_2$ more reflects the $SO_2$ emission rate of the region; (3) because $NH_4NO_3$ is much stable and much less affected by ambient temperature, $RNO_3$ is mainly determined by the availability of $NH_3$ over the region.

Based on the spatial patterns of air concentration of $NO_3^-$ and $SO_2$, and $RNO_3$ during cold season of

1989-1991, which are shown in Table S1.b and Fig. S1.b, we clustered the EUS and EC into 4 regions:





Region 1: sites located north of latitude 40º, and with concentration of $SO_2$ less than 6.4 µg m$^{-3}$ in cold season.

Region 2: sites with mean concentration of $NO_3^-$ greater than 2.5 µg m$^{-3}$. Except for site ARE128 at 2.1 µg m$^{-3}$, the highest air concentration of $NO_3^-$ of all other sites is 1.9 µg m$^{-3}$. For sites in region 2, $RNO_3$ was greater than 54%, which was higher than any CASTNET and CAPMoN sites in other regions.

Region 3: sites excluded from regions 1 and 2, and with an air concentration of $SO_2$ greater than 15.0 µg m$^{-3}$ during cold season.

Region 4: all other sites excluded from Regions 1, 2 and 3. The highest mean $SO_2$ of sites in region 4 during cold season was less than 11.7 µg m$^{-3}$.

Clustering of sites is shown in Fig. 1 with site names. Regions 2, 3, and 4 roughly correspond to the Midwest, Mid-Atlantic, and Southeast regions participating in the NBP (Butler et al., 2011). Characteristics of each region are listed in Table 1.

## 3. Results and discussions

### 3.1 Air quality in the eastern US and Eastern Canada at the beginning of the study period: 1989-1991

Three-year average from 1989-1991 was used to describe the air quality at the beginning of the study period, and was shown in Table S1.a and b for warm and cold seasons. Mapping of 3-year average for each species is also provided in Fig. S1 as supplement material for this paper. First of all, among 4 regions and both warm and cold seasons, region 1 had the lowest air concentration for all species with mean $NO_3^-$, $HNO_3$ and $NH_4^+$ concentrations less than 1.0 µg m$^{-3}$. The mean air concentration of $NO_3^-$ during warm season was only 0.14 µg m$^{-3}$. Mean $SO_4^{2-}$ concentrations were 2.9 and 2.3 µg m$^{-3}$ during warm and cold seasons respectively, and $SO_2$ was 1.6 and 3.6 µg m$^{-3}$ correspondingly.



For regions 2-4, $SO_4^{2-}$ was highest in region 3 and lowest in region 4 for both seasons, varying from 7.6 to 8.2 µg m$^{-3}$ during warm season and 3.6 to 4.2 µg m$^{-3}$ during cold season. The difference in $SO_4^{2-}$ between region 2 and 3 was less than 0.1 µg m$^{-3}$ during cold season. Generally $SO_4^{2-}$ in regions 2-4 was relatively uniform. For each region $SO_4^{2-}$ during warm season was about double of that during cold

season. Same as $SO_4^{2-}$, $SO_2$ was also highest in region 3 and lowest in region 4 for regions 2-4, but $SO_2$ in region 3 was much higher and was about 2.5 times of that in region 4. $SO_2$ in region 3 and 2 during cold season, being 19.2 and 13.7 µg m$^{-3}$ respectively, were the two highest concentrations and the only two concentrations greater than 10.0 µg m$^{-3}$ among all species in 4 regions and during the warm and cold seasons. Difference in $SO_2$ between region 3 and 2 was less than 1.0 µg m$^{-3}$ during warm

season, but was more than 5.0 µg m$^{-3}$ during cold season. Despite significant differences in $SO_2$ for regions 2-4, the corresponding differences in $SO_4^{2-}$ were pretty small. As an example, during cold season, $SO_2$ in region 3 was larger than that in region 4 by 10.0 µg m$^{-3}$, but the corresponding difference in $SO_4^{2-}$ was only 0.7 µg m$^{-3}$. This can be attributed to the fact that the life time of SO4 in the air is much longer than that of SO2 (references). Comparing $SO_2$ in cold season to that in warm

season, it was about 2 times larger in region 3 and 4, and 59% larger in region 2. In contrast to the pattern of $SO_4^{2-}$, $NO_3^-$ in regions 2-4 was significantly different from region to region. Region 2 had the highest concentration of $NO_3^-$ at 1.5 µg m$^{-3}$ during warm season and 3.6 µg m$^{-3}$ during cold season, which were about triple of the second highest value in region 3. Region 4 had the lowest $NO_3^-$ concentrations among regions 2-4, being 0.3 and 0.6 µg m$^{-3}$ for warm and cold seasons respectively.

The lowest value for stations in region 2 during cold season was 2.5 µg m$^{-3}$ at ALH157, higher than the highest value of 2.1 µg m$^{-3}$ at BEL116 in region 3. During warm season, HNO$_3$ ranged from 1.7 µg m$^{-3}$ in region 4 to 2.8 and 2.9 µg m$^{-3}$ in regions 3 and 2. During cold season, the highest concentration of HNO$_3$ was in region 3, and the lowest in region 2, being 2.3 and 1.8 µg m$^{-3}$ respectively. Considering both seasons, region 3 had the highest concentration of HNO$_3$ among 4 regions. Region 2 had the

lowest concentration of HNO$_3$ among regions 2-4 during cold season due to that significant portion of





HNO$_3$ was neutralized by NH$_3$ to form NH$_4$NO$_3$. For TNO$_3$, in both seasons, region 2 had the highest concentration, being 4.3 and 5.3 µg m$^{-3}$ in warm and cold seasons, mainly because of the significantly higher concentration of NO$_3^-$ than other regions. TNO$_3$ was 3.2 and 3.5 µg m$^{-3}$ for region 3, 2.0 and 2.3 µg m$^{-3}$ for region 4 during warm and cold seasons. NH$_4^+$ in regions 2-4 varied from 1.9 to 2.7 µg m$^{-3}$

during warm season, and 1.1 to 2.3 µg m$^{-3}$ during cold season, with the highest concentrations in region 2 and lowest concentrations in region 4 for both seasons. NH$_4^+$ was higher in warm season than in cold season for all regions, as much more (NH$_4$)$_2$SO$_4$ formed in warm season than in cold season.

In general, region 1 had the lowest concentration of all species among the 4 regions, and region 4 had the second lowest except HNO$_3$ was a little bit more than that in region 2 during cold season. Region 2

and 3 were the two most polluted regions in EUS and ECA. Region 3 had the highest concentration of SO$_2$ in both seasons, more than double of that in region 1 and 4; Region 2 had the highest concentration of NO$_3^-$ and TNO$_3$. In both seasons, NO$_3^-$ in region 2 was more than 4 times larger than that in regions 1 and 4, and TNO$_3$ was more than double of that in regions 1 and 4.

## 3.2 Time series of annual concentration of SO$_4^{2-}$, SO$_2$, NO$_3^-$, HNO$_3$, TNO$_3$ and NH$_4^+$ during 1989-2016

Time series of annual concentrations of SO$_4^{2-}$, SO$_2$, NO$_3^-$, HNO$_3$, TNO$_3$ and NH$_4^+$ are shown in Fig. 2.a for warm season and Fig. 2.b for cold season. Time series of regional averages for regions 1-4 normalized to year 2000 are presented in Fig. 3.a and 3.b for warm and cold seasons. As shown in Fig. 2, the time

series for stations within each region have pretty high correlations after the stations are properly grouped into 4 regions.

3.2.1   SO$_4^{2-}$ and SO$_2$

3.2.1.1 SO$_4^{2-}$ and SO$_2$ during cold season



$SO_4^{2-}$ in region 1 had a steady decreasing trend from 1989 to 2000, then a relatively slower decreasing trend from 2001 to 2007. There was a big drop from 2008 to 2010, followed by a slow decreasing period of 2011-2016. During 1989-2016, there were 3 relatively significant increases in 2001, 2008, and 2011. $SO_4^{2-}$ in region 2 decreased significantly during 1989-1990 and 1993-1995, and was

relatively steady during 1990-1993 and 1995-1998. The trend for 1998-2005 was generally steady decreasing with peaks in 2001, 2003, and 2005, and minima in 1999, 2002 and 2004. From 2005 to 2016, there was a straight decreasing trend with small increases in 2008 and 2014. $SO_4^{2-}$ in region 3 decreased significantly in 1989-1991, 1993-1994, and 1996-1999, and was relatively steady in 1991-1993 and 1994-1996. In general, $SO_4^{2-}$ decreased significantly for the period 1989-1999. There was an

obvious increasing in 1999-2001, and then $SO_4^{2-}$ decreased slowly with peaks in 2003, 2005 for the period 2001-2008. Excluding the initial increase and subsequent decrease, the change of $SO_4^{2-}$ during 1999-2008 was very small. There was a smooth decreasing trend from 2008 to 2016, and the drop of the annual concentration was significant in 2008-2011 and 2014-2016. $SO_4^{2-}$ in region 4 had a peak in 1989 for 1988-1990, which was followed by three steady periods of 1990-1993, 1994-2001 and 2002-

2007 with averages of $SO_4^{2-}$ for the periods being 3.4 µg m$^{-3}$, 3.1 µg m$^{-3}$, and 2.8 µg m$^{-3}$ respectively. There were only two major drops between the periods, in 1994 and 2002 respectively. From 2007 to 2016, $SO_4^{2-}$ in region 4 had a steep drop, and the concentration was reduced from 2.7 µg m$^{-3}$ in 2007 to 1.1 µg m$^{-3}$ in 2016. In general, $SO_4^{2-}$ in EUS and EC during cold season can be described by two fast decreasing periods of 1989-1995 and 2005-2016, and a slow decreasing period of 1995-2005. The

annual reduction rates during the three periods were 0.14, 0.03, and 0.05 µg m$^{-3}$yr$^{-1}$ in region 1; 0.16, 0.04, and 0.13 µg m$^{-3}$yr$^{-1}$ in region 2; 0.15, 0.05, and 0.15 µg m$^{-3}$yr$^{-1}$ in region 3; and 0.10, 0.04 and 0.14 µg m$^{-3}$yr$^{-1}$ in region 4. The decreasing rates in regions 2 and 3 were pretty close. If all sites within regions 1-4 were combined, the corresponding rates were 0.14, 0.04, and 0.12 µg m$^{-3}$yr$^{-1}$ for the three periods.


$SO_2$ in regions 2-4 during cold season had a significant drop in 1989-1995 with a temporary increase in 1993 and 1994. From 1995 to 2005 the decreasing trend was pretty slow, and then there was a very steep reduction from 2005 to 2012. The trend from 2012 to 2016 was relatively flat. The trend of $SO_2$ in region 1 was similar except that there was no obvious increase in 1993- 1994.

$SO_2$ achieved an annual reduction rate of 0.28, 0.06, and 0.12 $\mu g\ m^{-3}yr^{-1}$ in region 1;  0.83, 0.09, and 0.73 $\mu g\ m^{-3}yr^{-1}$ in region 2; 1.13, 0.22, 1.13 $\mu g\ m^{-3}yr^{-1}$ in region 3; 0.32, 0.08, and 0.49 $\mu g\ m^{-3}yr^{-1}$ in region 4 for the periods of 1989-1995, 1995-2005, and 2005-2012. The annual reduction rate in region 1, 2 and 4 was less than 0.1 $\mu g\ m^{-3}yr^{-1}$ for 1995-2005.  For the whole regions 1-4, the corresponding reduction rates were 0.72, 0.16 and 0.48 $\mu g\ m^{-3}yr^{-1}$. For the third period, if we only consider 2005-

2012, the annual reduction rate was 0.61 $\mu g\ m^{-3}yr^{-1}$. Although the trend of $SO_2$ during 2012-2016 was generally flat, the decreasing of $SO_4^{2-}$ during the period was still significantly, especially in terms of percentage.

3.2.1.2  $SO_4^{2-}$ and $SO_2$ during warm season

$SO_4^{2-}$ in region 1 had a significant decrease from 1989 to 1996, and it was followed by a level off until

2005. There was a decreasing trend in 2005-2016 with a small peak in 2007. After a significant peak in 1989, $SO_4^{2-}$ in region 2 had a drop from 1989 to 1993 and an increase in 1994. There was a steady decreasing trend for 1994-2004, followed by a significant peak in 2005 which was captured by all measurement sites within the region. There was a smooth decreasing trend for 2005-2016, with significant drops in 2005-2006, 2007-2009, 2012, and 2013-2016. Unlike region 2, $SO_4^{2-}$ in region 3 had

no significant peak in 1989, and had a decreasing trend during 1989-1999 with peaks in 1991, 1994, and 1998. There was a level off for 1999-2005, and it was followed by a significant decreasing trend for 2005-2016 with steep drops in 2005-2006, 2007-2009, and 2011-2016. $SO_4^{2-}$ in region 4 had a decreasing trend in 1989-1994 with a peak in 1993, a slow decreasing trend in 1994-2000 with a peak in 1998, a drop in 2001, and another level off in 2001-2007 with a peak in 2005. There was a significant



straight decreasing trend for 2007-2016 with a very significant drop during 2007-2009. $SO_4^{2-}$ in all stations converged to the regional average after 2009. In general, the trend of $SO_4^{2-}$ during warm season can be characterized by two fast reduction periods of 1989-1995 and 2007-2016, and a slow period 1995-2007. For the three periods, $SO_4^{2-}$ had a reduction rate of 0.17, 0.01 and 0.14 µg m$^{-3}$yr$^{-1}$ for region 1; 0.39, 0.03 and 0.43 µg m$^{-3}$yr$^{-1}$ for region 2; 0.28, 0.01 and 0.52 µg m$^{-3}$yr$^{-1}$ for region 3; and 0.24, 0.04 and 0.50 µg m$^{-3}$yr$^{-1}$ for region 4.

$SO_2$ during warm season was pretty low during 1990-2016 in region 1. There was a slow decreasing trend from 1990 to 2007, a relatively large decreasing in 2007-2009, and a level-off at pretty low concentration ( < 1.0 µg m$^{-3}$) in 2009-2016. $SO_2$ during warm season in regions 2-4 had similar trends: (1) a fast decreasing period of 1989-1995, with a level off period of 1992-1994 in region 2 and a peak in 1993 in regions 3 and 4; (2) a slowing decreasing period of 1995-2007 with an initial slow increase in 1995-1999 in region 2, and an initial significant increase in 1996-1998 in region 3 and 4; (3) a fast decreasing period of 2007-2016 with very steep decrease in 2007-2009. $SO_2$ had an annual reduction rate of 0.12, 0.03 and 0.04 µg m$^{-3}$yr$^{-1}$ for region 1; 0.55, 0.08, and 0.43 µg m$^{-3}$yr$^{-1}$ for region 2; 0.46, 0.11 and 0.48 µg m$^{-3}$yr$^{-1}$ for region 3; 0.13, 0.05 and 0.22 µg m$^{-3}$yr$^{-1}$ for region 4. For the whole region, the annual reduction rates during warm season were 0.31, 0.04 and 0.38 µg m$^{-3}$yr$^{-1}$ for $SO_4^{2-}$, and 0.34, 0.08, and 0.30 µg m$^{-3}$yr$^{-1}$ for $SO_2$ for the periods of 1989-1995, 1995-2007, and 2007-2016.

### 3.2.2 $NO_3^-$, $HNO_3$ and $TNO_3$

#### 3.2.2.1 $NO_3^-$, $HNO_3$ and $TNO_3$ during cold season

$NO_3^-$ in EUS and EC during cold season was dominated by $NO_3^-$ in region 2, which was much higher than $NO_3^-$ in other regions. $NO_3^-$ in region 2 had no obvious trend in the period of 1989-2001, but there



was a decreasing trend for the period 2001-2016. The trends of $NO_3^-$ in regions 1, 3 and 4 over the whole study period were pretty flat.

Excluding a peak of 1.9 $\mu g\ m^{-3}$ in 1993, the trend of $HNO_3$ in region 2 in 1990s is pretty flat. From 2003 to 2013, there was a decreasing trend, then a slow increasing trend for 2013-2016 in region 2.

Excluding two minima in $HNO_3$ concentration during 1991-1992, and 1998-1999, $HNO_3$ in region 3 went through a general declining trend by 21.7% (2004-2005 vs. 1989-1990) during 1989-2005. The declining trend of $HNO_3$ in 1990s was consistent with the increasing trend of $NO_3^-$ during the period, as more $HNO_3$ was neutralized by more excessive $NH_3$ released from decreasing $SO_4^{2-}$. $HNO_3$ decreased significantly from 2005 to 2009 in region 3, then went through a slow decreasing trend for 2009-2016.

$HNO_3$ in region 4 had an initial decrease from 1989 to 1991, then an increasing trend of 21.1% (1999-2000 vs. 1991-1992) for 1991-2000. The region went through a slow decreasing for 2000-2005, a fast decreasing for 2005-2009, and a slow decreasing for 2009-2016.

$TNO_3$ in cold seasons over region 2, 3 and 4 had similar trends. It decreased from 1989 to 1992, then increased until 1995, decreased again until 1998, and peaked again in 2001 (for region 4, peaked at

2000-2001). Excluding the up-and-down cycles, the general trend of $TNO_3$ over region 2, 3, and 4 was from steady to slow increasing for 1990-2001. This is consistent with the $NO_x$ emission trend during the period. For the period 2001-2016, there was significant decreasing of $TNO_3$ for the three regions. For region 3, it happened mainly during the period 2001-2009.

3.2.2.2 $NO_3^-$, $HNO_3$ and $TNO_3$ during warm season

$NO_3^-$ in region 1 had an increasing trend in 1990-2000, a decreasing trend in 2000-2007, and a flat trend in 2007-2016. Regression of annual $NO_3^-$ concentration in region 2 during warm season from 1990 to 1999 shows a slow increasing trend for $NO_3^-$ in 1990s. There were two peaks in 1989 and 2000, and two big drops in 1990 and 2001. After 2002, there was a straight decline of $NO_3^-$ concentration until 2009, then followed by a small increasing during 2009-2011 and a steady



decreasing from 2011-2016. Regression of $NO_3^-$ in region 3 shows a flat trend during 1989-2000 and a weak increasing trend during 1990-1999. Between 2000 and 2005, $NO_3^-$ in region 3 went through a clearly declining trend. From 2005 to 2016, the trend is generally flat. Annual concentration of $NO_3^-$ in region 4 peaked at 0.38 and 0.39 $\mu g\ m^{-3}$ in 1989 and 1991, bottomed at 0.21 and 0.23 $\mu g\ m^{-3}$ in 1990

and 1992, and peaked again in 2001 at 0.37 $\mu g\ m^{-3}$. There was a weak increasing trend for 1992-2000, followed by a decreasing trend for 2000-2005 and a slow increasing trend for 2005-2016. In general, $NO_3^-$ during warm season had a flat to weak increasing trend for 1990-2000 and a strong decreasing trend for 2000-2005 for regions 2-4. For 2005-2016, there was a decreasing trend for region 2, a flat trend for region 3, and a weak increasing trend for region 4.

$HNO_3$ in region 1 was pretty low ( < 1.0 $\mu g\ m^{-3}$) in general during the whole study period. There was flat trend in 1990 to 2001, a slow decreasing trend in 2001 to 2009, and a flat trend again in 2009 to 2016. $HNO_3$ in regions 2-4 can be characterized by three periods: a flat (regions 2 and 3) or a week increasing (region 4) trend for 1989-1999, a significant decreasing trend for 1999-2009, and a generally flat trend for 2009-2016.

$HNO_3$ dominated over $NO_3^-$ in $TNO_3$ during warm season for all regions, and especially in regions 3 and 4, where ratio of $NO_3^-$ to $TNO_3$ usually was less than 20%. Therefore, the trend of $TNO_3$ generally followed that of $HNO_3$ during warm season. In 1990s (1990-1999) the trend of $TNO_3$ in warm season over regions 2 and 3 was pretty flat, and there was a very weak increasing trend over region 4. The turning point of the trend was in 1999. For the period 1999-2009, all three regions went through

significant decreasing trend. For 2009-2016, the trends in the three regions were generally flat.

3.2.3 $NH_4^+$

3.2.3.1 $NH_4^+$ during cold season



NH$_4^+$ during cold season in region 1 had a flat trend in 1990-1999, a very slow decreasing trend in 1999-2009, and a general flat trend in 2009-2015. The trend of NH$_4^+$ in region 2 during cold season was affected by the trends of both SO$_4^{2-}$ and NO$_3^-$. Following that of SO$_4^{2-}$, NH$_4^+$ in region 2 was generally decreasing for 1989-1999, but at a much less significant rate. From 1993 to 1995, there was a

significant decrease of SO$_4^{2-}$, but NH$_4^+$ changed very little, as at the same period NO$_3^-$ was increased by more than 25%. The peaks of the trend in 2001 and 2005 followed those of SO$_4^{2-}$ and NO$_3^-$. After 2001, the trend of NH$_4^+$ followed that of SO$_4^{2-}$ better, as NO$_3^-$ also went through a decreasing trend. NH$_4^+$ in region 3 had an initial drop in 1990 as region 2, then went through a weak decreasing trend until there was an increasing in 1995 due to the significant increase of NO$_3^-$. NH$_4^+$ had a decreasing trend for

1996-1999. The increase in 1999-2001 was consistent with that of SO$_4^{2-}$ and NO$_3^-$ during the period. From 2001-2016, NH$_4^+$ went through a similar decreasing trend as SO$_4^{2-}$. NH$_4^+$ in region 4 had a drop in 1989-1990, then remained unchanged during 1990-2001. After 2001, NH$_4^+$ decreased steadily until 2016.

### 3.3.3.2 NH$_4^+$ during warm season

NH$_4^+$ in region 1 changed little in 1990-2005 and 2009-2015 during warm season, but there was a decreasing trend in 2005-2009. NH$_4^+$ in region 2 had two significant peaks during 1989-2011, in 1989 and 2005 respectively. The two peaks can be seen at all measurement sites within the region. The peak in 2005 was associated with SO$_4^{2-}$ only and not with NO$_3^-$. Excluding the two significant peaks, NH$_4^+$ went through a consistent and steady decreasing trend from 1990 to 2016. Fig. 3 shows that the

trend of NH$_4^+$ in region 3 generally followed that of SO$_4^{2-}$ very well because NH$_4^+$ in region 3 was dominantly associated with SO$_4^{2-}$ during warm season. The two trends were almost identical after 2000, but the decrease of NH$_4^+$ before 2000 was less significant than that of SO$_4^{2-}$, as part of NH$_4^+$ was associated with NO$_3^-$ and NO$_3^-$ didn't have a declining trend for 1990-2000. From 1995-2005, the trend of NH$_4^+$ in region 3 was almost flat as during this period SO$_4^{2-}$ didn't decrease significantly. The trend

from 2005 to 2011 was identical to that of SO$_4^{2-}$. There was a very significant decrease of NH$_4^+$ in



region 3 in 2005-2009 and it decreased by almost 50% in 4 years. This was due to that $SO_2$ decreased by more than 50% during the same period in region 3. $NH_4^+$ in region 4 changed little from 1990 to 2000, while at the same period $SO_4^{2-}$ had an obvious decreasing trend. This is because $NO_3^-$ had an increasing trend during this period. After 2000 the trend of $NH_4^+$ followed $SO_4^{2-}$ very well: there was little change in 2001-2007 except there was a peak in 2005, a dramatic decrease in 2007-2009 by 45%, a small increase in 2009-2010, and a steady decreasing trend for 2010-2016.

## 3.3 10 and 25 years of changes of $SO_4^{2-}$, $SO_2$, $NH_4^+$, $NO_3^-$, $HNO_3$ and $TNO_3$ in EUS and EC for 1990-2015

10 and 25 years of changes of $SO_4^{2-}$, $SO_2$, $NH_4^+$, $NO_3^-$, $HNO_3$, and $TNO_3$ during 1990-2015 are presented in Tables 2 and 3, and are summarized in sections 3.3.1 and 3.3.2. The changes are calculated with 3-year average centered at 1990, 2000, and 2015.

### 3.3.1   10 years of changes for the period 1990-2000

1. During the first 10 years, $SO_2$ was significantly reduced in all regions and seasons by more than 25.0% except region 4, which had a reduction of 15.5% during warm season and 23.8% during cold season.

2. $SO_4^{2-}$ went through a similar but less significant decreasing trend as $SO_2$. The reduction was more than 20% except region 4 during cold season. Region 4 during warm season had a similar reduction rate of $SO_4^{2-}$ as region 3 despite the significant difference in the reduction rate of $SO_2$ between the two regions.

3. $NO_3^-$ increased between 6.6% and 40.0% during cold season for regions 1-4. Changes of $NO_3^-$ during warm season in region 3 and 4 were very small, and only had a significant reduction of 9.6% in region 2.





4. $TNO_3$ increased little in region 1, by 0.09 and 0.02 $\mu g\ m^{-3}$ for cold and warm seasons respectively. $TNO_3$ in regions 2 and 3 changed very little during cold season, and had a 9.4% and 11.8% reduction during warm season. $TNO_3$ in region 4 increased by 3.9% during warm season, and by 14.2% during cold season.

5. $NH_4^+$ reduced by 12% to 29.8% during both seasons, except for a negligible change in region 4 during cold season.

6. In general, for the first ten-year period of 1990-2010, $SO_2$, $SO_4^{2-}$ and $NH_4^+$ reduced by 31.6%, 26.7% and 18.5% respectively in EUS and EC. $HNO_3$ reduced in regions 1-3. $NO_3^-$ increased in regions 1-4 during cold season, and changed very little (< 0.15 $\mu g\ m^{-3}$) during warm season. Considering both

seasons and all regions, $NO_3^-$ increased by 12.7%, $HNO_3$ reduced by 5.6%, and change of $TNO_3$ was negligible with mean concentration being 3.02 $\mu g\ m^{-3}$ for 1989-1990 vs. 3.05 $\mu g\ m^{-3}$ for 1999-2001.

**3.3.2 25 years of changes for the period 1990-2015**

During the 25-year period of 1990-2005, air quality in EUS and EC went through very significant

changes, and are summarized as following:

1. Among all species, the most significant reduction during the period was $SO_2$. Reduction of $SO_2$ in regions 2-4 was similar in percentage, from 83.9% in warm season for region 4 to 91.2% in warm season for region 3. There was no obvious difference between warm and cold seasons in terms of percentage of reduction. In terms of absolute value, the biggest reduction was $SO_2$ in

region 3 during cold season, and the mean annual concentration was reduced from 19.2 $\mu g\ m^{-3}$ to 2.2 $\mu g\ m^{-3}$.

2. Change of $SO_4^{2-}$ during cold season was relatively uniform in term of percentage, ranging from 60.1% in region 2 to 62.5% in region 3, and it was more significant during warm season, ranging

from 72.7% in region 1 to 78.7% in region 4. Changes of $SO_4^{2-}$ in regions 2, 3, and 4 were similar in terms of absolute concentrations and percentages, and in both seasons. Reduction of $SO_4^{2-}$ in terms of percentage was much smaller than $SO_2$ in all regions during both seasons except $SO_4^{2-}$ during warm season in region 1.

3. During warm season, reduction of $NO_3^-$ was seen in 4 regions, ranging from 14.3% and 15.6% in regions 1 and 4, to 36.2% and 57.5% in regions 3 and 2. Reduction of $NO_3^-$ during cold season was only observed in region 2 by 30.4%. Changes of $NO_3^-$ in region 3 and 4 during cold season were negligible. Although $TNO_3$ was reduced during cold season in regions 3 and 4, a higher percentage of $HNO_3$ was converted to $NO_3^-$ as more excessive $NH_3$ was available to form $NH_4NO_3$ due to the reduction of $SO_4^{2-}$. Therefore, the trend of $NO_3^-$ in the two regions during cold season changed very little. Unlike regions 3 and 4, region 2 did observe a significant reduction of $NO_3^-$ in cold season, following a 38.3% reduction of $TNO_3$. This can be explained as region 2 is an $NH_3$-ample region. Formation of $NH_4NO_3$ during cold season in the region is less sensitive to the excessive $NH_3$ released from $SO_4^{2-}$ reduction than in regions 3 and 4. This can also be demonstrated by the least reduction of $HNO_3$ (in terms of percentage) in region 2 during cold season.

4. Reduction of $HNO_3$ was similar among four regions during warm season, from 63.1% to 68.8%. During cold season, region 2 had the lowest percentage of reduction at 56.0%, and region 1 had the highest at 63.5%. Reduction of $HNO_3$ can be through two paths: reduction of $NO_x$ emission and increased neutralization of $HNO_3$ by more excessive $NH_3$ released from $SO_4^{2-}$ reduction. Reduction of $HNO_3$ was more significant than $TNO_3$ during cold season, ranging from 14.4% more in region 4 to 28.0% more in region 1.

5. $TNO_3$ had a reduction ranging from 35.5% for cold season in region 1, to 64% during warm season in region 3. The reduction during warm season was much higher than in cold season, ranging from 11.4% higher in region 4 to 23.9% higher in region 3. The difference was due to extra reduction of $NO_x$ emission during warm season after 1999 (Butler et al. 2011).



6. Reduction of $NH_4^+$ was similar in regions 2, 3, and 4, ranging from 48.9% to 53.2% in cold season, and from 74.0% to 75.7% in warm season. Reduction of $NH_4^+$ during warm season was more significant than in cold season, over 20% more in regions 2-4. The reduction of $NH_4^+$ generally followed the trend of $SO_4^{2-}$, but the reduction rate was much lower than that of $SO_4^{2-}$ during cold season because more percentage of $NH_4^+$ was associated with $NO_3^-$ and reduction of $NO_3^-$ was not as significant as $SO_4^{2-}$ during cold season. Region 2 achieved the largest reduction of $NH_4^+$ by 75.7% during warm season, due to a 76.8% reduction of $SO_4^{2-}$ as well as a 57.5% reduction of $NO_3^-$.

7. $RSO_4$ increased the most in region 3 during cold season at 166.3%, and the least in region 1 during warm season at 0.6%. During warm season, $RSO_4$ increased by more than 50% in regions 2 and 3, being 54.9% and 58.4% respectively. The increase of $RSO_4$ during cold season was much higher than in warm season in terms of percentage, ranging from 48.6% in region 1 to 166.3 % in region 3.

8. $RNO_3$ increased significantly in regions 1, 3, and 4 in both seasons, ranging from 73.9% to 94.9%, but $RNO_3$ only increased by 8.0% and 12.8% in warm and cold seasons for the $NH_3$-ample region 2.

9. As presented in Table 3, for the whole region: (1) among the 5 species of $SO_4^{2-}$, $NO_3^-$, $NH_4^+$, $HNO_3$ and $SO_2$, only regionally averaged $SO_4^{2-}$ and $SO_2$ still exceeded 1.0 $\mu g\ m^{-3}$, while at the beginning of the period, all 5 species exceeded this value; (2) $SO_4^{2-}$ was reduced by 73.3% annually for the whole region , and it was reduced about 15% more in warm season than in cold season in term of percentage; (3) $NH_4^+$ was reduced more in warm season than in cold season, in terms of both percentage and absolute value; (4) $NO_3^-$ was reduced by 29.1% for the whole region. The reduction of $NO_3^-$ during cold season occurred only in region 2, and the reduction during warm season occurred mostly in regions 2 and 3. The reduction for the whole region was mainly due to reduction of $NO_3^-$ in region 2 during warm and cold seasons; (5) $RSO_4$



increased by 97.7% in cold season, much higher than 26.2% in warm season. RSO₄ increased the most in region 2 during cold season, in both absolute value and percentage.

### 3.4 Air quality at the end of the study period: 2014-2016

3-year averages for 2014-2016 are used to describe the air quality at the end of the study period, and are presented in Table S2 and Fig. S2.

As at the beginning of the period, region 1 had the cleanest air among all regions with the lowest air concentration, being less than 1.0 µg m⁻³ for all species of both warm and cold seasons.

Unlike at the beginning of the period that $SO_4^{2-}$ during warm season was about double of that during
cold season in regions 2-4, $SO_4^{2-}$ at the end of the period had no significant differences between the two seasons. Air concentrations of $SO_4^{2-}$ were less than 2.0 µg m⁻³ in all regions and both seasons. For regions 2-4 the regional averages ranged from 1.6 to 1.8 µg m⁻³ during warm season, and from 1.4 to 1.7 µg m⁻³ during cold season. $SO_2$ during warm season was only from 0.6 to 1.0 µg m⁻³ for regions 2-4. In cold season, $SO_2$ in region 2 and 3 was the same at 2.2 µg m⁻³, and was only 1.1 µg m⁻³ in region 4.
$NH_4^+$ during warm season varied from 0.5 to 0.7 µg m⁻³ for regions 2-4. During cold season, it was 0.5 and 0.8 µg m⁻³ in regions 4 and 3 respectively, and it was much higher in region 2 at 1.2 µg m⁻³. $NO_3^-$ during warm season was very low in region 3 and 4 at 0.3 µg m⁻³, and it was doubled in region 2 at 0.6 µg m⁻³. During cold season, $NO_3^-$ was much higher than during warm season, being 2.5, 1.3 and 0.5 µg m⁻³ for regions 2, 3, and 4 respectively. $HNO_3$ in regions 2-4 varied from 0.6 µg m⁻³ in region 4 to 1.1 µg
m⁻³ in region 2 during warm season, and from 0.7 µg m⁻³ in region 4 to 0.9 µg m⁻³ in region 3 during cold season. There was little difference between warm and cold seasons in regions 3 and 4. Total $NO_3^-$ was highest in region 2 in both seasons, being 1.7 and 3.2 µg m⁻³ for warm and cold seasons





respectively. Region 3 had the second highest $TNO_3$ at 1.2 µg m$^{-3}$ and 2.1 µg m$^{-3}$ for warm and cold seasons, and the corresponding values for region 4 were 0.9 µg m$^{-3}$ and 1.3 µg m$^{-3}$.

In summary, for species of $SO_4^{2-}$, $NO_3^-$, $NH_4^+$, $HNO_3$ and $SO_2$, region 1 had air concentration of less than 1.0 µg m$^{-3}$ for all species in both seasons. For regions 2-4, $NO_3^-$ was less than 1.0 µg m$^{-3}$ for all regions and both seasons except region 2 and 3 during cold season at 2.5 and 1.3 µg m$^{-3}$; $HNO_3$ was less than 1.0 µg m$^{-3}$ except region 2 during warm season at 1.1 µg m$^{-3}$; $NH_4^+$ was less than 1.0 µg m$^{-3}$ for all regions except region 2 during cold season at 1.2 µg m$^{-3}$; $SO_4^{2-}$ was greater than 1.0 but less than 2.0 µg m$^{-3}$ for regions 2-4 and both seasons; $SO_2$ was greater than 1.0 but less than 2.5 µg m$^{-3}$ for regions 2-4 and both seasons, except regions 3 and 4 during warm season at 0.8 and 0.6 µg m$^{-3}$. Among 4 regions, region 2 had the highest air concentration for all species except $HNO_3$ during cold season. Especially $NO_3^-$ in region 2 was double of the second highest value in region 3 in both seasons. Also $NO_3^-$ in region 2 was the highest in value ( at 2.5 µg m$^{-3}$) among all species in 4 regions and both seasons, although it significantly dreased from 3.6 µg m$^{-3}$ at the beginning of the study period.

## 4. Long-term trends derived with polynomial regressions

Through trial and error, we found that polynomial regressions can reasonably describe the long-term trends of species for the period. Through these regressions, we can eliminate the relative-short-term variations due to meteorology. 4$^{th}$ order polynomial regressions were applied to normalized annual means of $SO_4^{2-}$, $SO_2$ and $NH_4^+$ during cold and warm seasons. For $NO_3^-$, $HNO_3$ and $TNO_3$, we applied 5$^{th}$ polynomial regressions to better capture the trends. Examples of regression for $SO_4^{2-}$ and $SO_2$ in region 3 during cold season are shown in Fig. 4. Fig. 5 shows the comparisons between normalized annual means during cold season in region 3 and the corresponding regressed values.



The regressed trends for $SO_4^{2-}$, $SO_2$, $NH_4^+$, $NO_3^-$, $HNO_3$ and $TNO_3$ for 4 regions during cold and warm seasons are shown in Fig. 6.a. The regressed trends are normalized to the regressed value of year 2000 as this is the turning point for the trend of $NO_3^-$. The regressed trends for regions 1-4 show clearly: (1) the most significant reduction of all species was $SO_2$; (2) There were significant disparities of reduction rate between $SO_4^{2-}$ and $SO_2$ during cold season. There were also disparities during warm season in regions 1-3, but much less significant than cold season; (3) the least significant reduction was $NO_3^-$ during the period. Fig 6.b show the regressed trends for each species during cold and warm seasons for different regions. For $SO_4^{2-}$ and $SO_2$, there were large differences for trends of 1989-2000 among between 1 and 4. Trends of $NH_4^+$ were different for regions for the period 1989-2000 during both seasons, and for the period 2000-2016 during cold season. During warm season, trends for $NO_3^-$ were similar for regions 2-4 for 1989-2000, but were different for 2000-2016. Trends of $HNO_3$ and $TNO_3$ were different for 1989-2000 during warm season, but were pretty similar for 2000-2016.

## 5. Discussions

### 5.1 RSO₄ and RSO₄ vs. SO₂

RSO4 is a metric describing how much sulfur in the air is oxidized from gas $SO_2$ to particle $SO_4^{2-}$. Similar metric, the ratio of $SO_2$ mass oxidized below 2 km to $SO_2$ mass emitted, was used by Shah et al (2018). RSO2, which is 1-RSO4, was used by Sickles II and Shadwick (2015). RSO4 is also an indicator of gas-particle partition ratio for sulfur in the air as $SO_2$ and $SO_4^{2-}$ exist in the air as gas and particle respectively. This metrics depends on a number of factors: the oxidation capacity of the air, the local emission rate of $SO_2$, the transportation of $SO_2$ and $SO_4^{2-}$ from upwind regions, and the time it takes to bring upwind $SO_2$ and $SO_4^{2-}$ to a local site. Generally the longer it takes to transport an upwind air particle, the more percentage of $SO_2$ is oxidized into $SO_4^{2-}$ and the more $SO_2$ is dry-deposited than $SO_4^{2-}$, so the higher RSO4; also the less percentage of $SO_2$ is emitted locally and the higher the





atmospheric oxidation capacity, the higher $RSO_4$. $RSO_4$ for 1989-1991 and 2014-2016, and the changes of $RSO_4$ between the two periods during warm and cold seasons are presented in Fig. S3. During warm season, because more solar photons are available to produce $O_3$ from $NO_x$ and VOCs, $RSO_4$ was much higher in warm season than cold season. At the beginning of the period, $RSO_4$ in warm season was

about double of that for cold season in regions 1, 2 and 4, and around triple in region 3. $RSO_4$ was much higher in regions 4 and 1 than in regions 2 and 3 because the local emission of $SO_2$ was much higher in regions 2 and 3. The "freshly" emitted $SO_2$ made $RSO_4$ in regions 2 and 3 relatively smaller. $RSO_4$ in region 3 during cold season was only 13.1% for 1989-1991, indicating a very low sulfur gas-particle partitioning ratio. Fig. S3 shows that during cold season, $RSO_4$ increased by more than 40% at

all sites except VPI120, which increased by 18.7%. During warm season, $RSO_4$ increased at all sites except VIP120, ASH135 and WST109, which decreased by 25.0%, 12.5% and 3.9% respectively. The most significant increase of $RSO_4$ was in region 3 during cold season, with a regional average of 166.3%.

Fig. 7 shows that $RSO_4$ increased with year linearly for region 1 and quadratically for regions 2-4 for

both seasons. $RSO_4$ increased significantly after 2005 in regions 2-4. Fig. 8 shows the correlations of $RSO_4$ vs. $SO_2$ for regions 2-4 and it clearly demonstrated that $RSO_4$ increased with decreasing of $SO_2$. The increase of $RSO_4$ was relatively slow when the concentration of $SO_2$ was greater than 5 $\mu g\ m^{-3}$ during cold season, and 7.5 $\mu g\ m^{-3}$ during warm season. $RSO_4$ soared when $SO_2$ was less than 5 $\mu g\ m^{-3}$ during cold season in regions 2-4, and less than 3 $\mu g\ m^{-3}$ during warm season in regions 2 and 3. The

increase of $RSO_4$ with decrease of $SO_2$ can be explained by the following: (1) the atmospheric oxidants didn't decrease as much as $SO_2$ emission. For example, the daily maximum 8 hour average $O_3$ only decreased by 14% for EUS during the May-September ozone season from 1997 to 2008 (Butler et al., 2011), and decreased by 4-15% during 1997-2006 for region 2, 3, and 4 (Chan, 2009). No significant decreasing trend was found for EUS during cold season for 1997-2006 (Chan, 2009). Sickles and

Shadwick (2015) found that $O_3$ in the eastern US decreased little during warm season and even





increased during cold season for 1990-2010. $O_3$ is an atmospheric oxidant, and is precursor to form other atmospheric oxidants, such as OH and $H_2O_2$. Therefore, relative to the significantly reduced $SO_2$, more atmospheric oxidants were available to oxidize $SO_2$, and $RSO_4$ increased significantly during the period. (2) $NH_3$ was relatively unchanged during the period, and even increased in some regions (Yao and Zhang, 2016). Decrease of $SO_2$ caused decrease of $H_2SO_4$. Together this made cloud or rain droplets or snow particles less acid, which was beneficial to oxidation of $SO_2$ by $H_2O_2$ in aqueous phase (Makar et al. 2009; Jones and Harrison, 2011).

Disparity of reduction of $SO_4^{2-}$ and $SO_4^{2-}$ in responses to emission reduction of $SO_2$, namely the reduction rate of $SO_2$ was faster than $SO_4^{2-}$, has been reported and discussed in some previous studies (Lovblad et al., 2004; Reid et al. 2001; Sickles II and Shadwick, 2015; Shah et al., 2018; Aas et al., 2019). The time series of normalized regional concentration of $SO_4^{2-}$ and $SO_2$ in Fig. 3 have clearly show the disparity during the period of 1990-2016. The significant increase of $RSO_4$ during the period, especially during cold season, explains why reduction rate of $SO_2$ was much higher than that of $SO_4^{2-}$ during the period, as reduction of $SO_2$ was due to not only the emission reduction, but also more percentage of $SO_2$ was converted to $SO_4^{2-}$. Faster reduction of $SO_2$ was observed for all 4 regions during cold season, both before and after year 2000, and it was more obvious after 2000. This can be explained by the fact that the increase of $RSO_4$ with time was very nonlinear. In the first 10 years of the study period, the increase of $RSO_4$ was relatively limited. During cold season, it was only increased by 7.3% in region 4 to 16.5% in region 3. It was in the last 10 years from 2005-2015 when $SO_2$ was further significantly reduced that $RSO_4$ increased dramatically. As an example, during cold season in region 3, $RSO_4$ was only increased by 16.5% in the first 10 years from 1990-2010, but it was increased by 149.8% for the last 15 years of the study period. During warm season, disparity of reduction between $SO_2$ and $SO_4^{2-}$ was much less, as clearly shown in Fig. 3. This is because the increase of $RSO_4$ during warm season was much less significant than during cold season (Table 4). In the first 10 years, $RSO_4$ was changed from - 4.1% in region 4 to 7.7% in region 1. For the period of 1990-2015, $RSO_4$ was only increased by 0.6%





and 12.4% percentage in regions 1 and 4. Disparity of reduction rate of $SO_2$ vs. $SO_4^{2-}$ for these two regions was only 1.7% and 5.2% during warm season respectively. This is expected. During warm season more atmospheric oxidants are produced due to more solar photons available than cold season, so oxidation of $SO_2$ is less limited by the availability of atmospheric oxidants during warm

season. During cold season, limited atmospheric oxidants are available for the oxidation of $SO_2$. Reduction of $SO_2$ in the air will make more atmospheric oxidants available to each $SO_2$ molecule, increase the oxidation rate of $SO_2$, and make $RSO_4$ increase.

## 5.2  Correlations of $SO_4^{2-}$ vs. $SO_2$

Correlations between $SO_4^{2-}$ and $SO_2$ are presented in Fig. 9 for regions 1-4 and for warm and cold seasons. The $SO_4^{2-}$-$SO_2$ relationship can be described by linear regressions for periods of 1990-2010 with R = 0.87 - 0.98 during warm season, and R = 0.96 - 0.99 during cold season. During cold season, region 1 had the highest slope, and it was followed by regions 4, 2, and 3. During warm season, the slopes for region 1 and 4 were similar, and were higher than slopes for region 2 and 3. A linear

relationship between $SO_4^{2-}$ and $SO_2$ indicates that there exists a linear relationship between $SO_4^{2-}$ and emission of $SO_2$. This is consistent with the relationship of $SO_4^{2-}$ concentration and $SO_2$ emission from the early 1990s through 2010 revealed in the study of Hand et al. (2012). As $RSO_4$ significantly increased when $SO_2$ was further reduced during 2010-2016, as seen in Fig. 8, the slopes of linear regression for 2010-2016 were much higher than those for 1989-2010. A power law regression, which

bends a linear regression with a gentle slope to a linear regression with a steep slope, described the $SO_4^{2-}$-$SO_2$ relationships very well with R= 0.97-0.98 during cold season, and R=0.94-0.99 during warm season, as shown in Fig. 10. In some previous studies (Jones and Harrison, 2011), non-linear power-law relationships have been found for observations at different sites and seasons, and for different



periods. Our results indicate that a linear relationship between $SO_4^{2-}$ and $SO_2$ exists for a sub-region of a long-term period, but generally the correlation of $SO_4^{2-}$ *vs.* $SO_2$ is a power-law relationship.

### 5.3 RNO₃

Similar to RSO₄ being a gas-particle partition indicator for sulfur in the air, RNO₃ is a metric indicating how much gas $HNO_3$ is aerosolized (Sickles and Shadwick, 2015). In the air, the emitted $NO_x$ is oxidized to gas $HNO_3$, which can be aerosolized through two paths: (1) reaction with $NH_3$ to form $NH_4NO_3$; (2) reaction with existing aerosols such as seal salts and crustal materials to form $NaNO3$, $Ca(NO_3)_2$, $Mg(NO_3)_2$ et al. The ratio is significantly sensitive to air temperature, as $NH_4NO_3$, $NH_3$, and $HNO_3$ in the
air are in equilibrium and temperature changes can affect the partitioning between gas and particle phases.

RNO₃ for 1989-1991 and 2014-2016, as well as the change of RNO₃ between the two periods are shown in Fig. S4. At the beginning of the period: (1) RNO₃ in cold season is much higher than warm season for all regions. RNO₃ of cold season in regions 2 and 3 was more than two times of that for
warm season; (2) RNO₃ in region 2 was much higher than other regions, and was more than double of RNO₃ in regions 3 and 4. For the 25 years of period, RNO₃ significantly increased by more than 70% in regions 1, 3 and 4 during both seasons. In region 2, RNO₃ only increased by 12.8% and 8.0% during cold and warm seasons respectively. The significant increase of RNO₃ in regions 1, 3, and 4 can be attributed to the significant reduction of $SO_4^{2-}$ during the period, as it is explained in the next section.

Fig. 7 shows that RNO₃ had an increasing trend with year for all regions and seasons except region 2 during warm season. The trends can be described well by linear regressions in regions 1 and 2, and by quadratic regressions in regions 3 and 4. RNO₃ in region 2 had a decreasing trend for 1990-2010 during warm season and the exact reason is unknown. One hypothesis is that due to global warming trend in





recent years, and significant reduction of sulfate and nitrate aerosols (which cool the atmosphere by reflecting more solar radiation back to space), near surface temperature in Midwest of US had an increasing trend during the period 1990-2010 (https://nca2014.globalchange.gov/report/regions/midwest/graphics/temperatures-are-rising-

midwest). As region 2 is ample of $NH_3$, $RNO_3$ is more sensitive to air temperature than to availability of $NH_3$. An increasing trend of air temperature in warm season can cause a decreasing trend of $RNO_3$.

### 5.4  Correlations of $RNO_3$ vs. $SO_4^{2-}$

Correlations between $RNO_3$ and $SO_4^{2-}$ for regions 2-4 and for warm and cold seasons are presented in

Fig. 10. For $NH_3$-ample region 2, $RNO_3$ increased slightly with decreasing of $SO_4^{2-}$ during cold season, and there was no obvious trend during warm season. For regions 3 and 4, which were $NH_3$-limited, $RNO_3$ increased with decreasing of $SO_4^{2-}$ concentration. Especially $RNO_3$ increased very significantly when $SO_4^{2-}$ was less than 4 $\mu g\ m^{-3}$ during warm season, and less than 3 $\mu g\ m^{-3}$ during cold season. The increase of $RNO_3$ with decreasing of $SO_4^{2-}$ can be explained as follows: In regions 3 and 4, formation of

$NH_4NO_3$ was limited by availability of $NH_3$. As $SO_4^{2-}$ decreases, part of $NH_3$ previously forming $(NH_4)_2SO_4$ was released, and was available to react with $HNO_3$ to form $NH_4NO_3$. In contrast, $RNO_3$ was much less sensitive to $SO_4^{2-}$ reduction in region 2 as emission of $NH_3$ there was much higher than that in regions 3 and 4, as seen in Fig. S5. Thus, in general there was always excess NH3 available to react with HNO3 to form $NH_4NO_3$ in region 2, which resulted in a lack of trends in RNO3. This also explained

while $TNO_3$ decreased by 40.1% and 46.4% respectively during cold season in region 3 and 4, the change of $NO_3^-$ was negligible in these two regions.

### 5.5  Correlations of $NH_4^+$ vs. $SO_4^{2-}$ and $NO_3^-$





Correlations of $NH_4^+$ vs. $SO_4^{2-}$ and $NH_4^+$ vs. $NO_3^-$ are shown in Fig. 11 for regions 2-4 and for cold and warm seasons. During warm season, $NO_3^-$ in regions 3 and 4 changed very little in value during 1989-2016, while $NH_4^+$ changed significantly and this change was dominantly and linearly associated with the change of $SO_4^{2-}$. In region 2, change of $NH_4^+$ was also dominantly associated with change of $SO_4^{2-}$,

but change of $NO_3^-$ also made a contribution to it. During cold season, $NH_4^+$ correlated with $SO_4^{2-}$ linearly very well in regions 2-4. In region 2, $NO_3^-$ also changed significantly during the period, and the variation of $NO_3^-$ correlated with variation of $NH_4^+$ relatively well, indicating part of reduction of $NH_4^+$ in region 2 was associated with reduction of $NO_3^-$ during the period. Variation of $NO_3^-$ in regions 3 and 4 were relatively small, and the correlations between $NH_4^+$ and $NO_3^-$ were much less significant than

those for $NH_4^+$ and $SO_4^{2-}$. Fig. 11 shows that in EUS and EC, reduction of $NH_4^+$ during 1989-2016 was mainly due to reduction of $SO_4^{2-}$ in regions 2-4, but in region 2 reduction of $NO_3^-$ also made a contribution to it.

### 5.6 Sulfate-nitrate-ammonium (SNA) aerosols

Sulfate, nitrate and ammonium are major components of secondary aerosols in the atmosphere. Time series of annual total mass of sulfate-nitrate-ammonium aerosols during warm and cold seasons are shown in Fig. 12. During cold season, mainly due to concentration of $NO_3^-$ and $NH_4^+$, SNA had the highest and second highest annual concentrations in region 4 and region 3 respectively, and the lowest annual concentration in region 1. During warm season, SNA in regions 2 and 3 was comparable,

and a little higher than that in region 4. The trends in regions 2-4 are pretty similar during warm season and the SNA in regions 2-4 are much higher than that in region 1. Fig. 13 shows that SNA in region 1 during warm season was higher than that during cold season until 2007, and the trend was reserved after that. In region 2, SNA was generally higher during warm season until 2005, and was opposite thereafter. SNA in region 3 was significantly higher during warm season than cold season

until 2007 and the trend was opposite after 2012. Similarly SNA in region 4 was much higher in warm season until 2008 and was pretty comparable between warm and cold seasons after 2012. Two points



can be derived the above trends: (1) in EUS and EC, SNA during warm season was mainly due to $(NH_4)_2SO_4$ / $NH_4HSO_4$. When emission of $SO_2$ over the region decreased significantly, SNA followed the decreasing trend even the reduction of $NO_3^-$ was not significant; (2) in EUS and EC, SNA during cold season was due to both $(NH_4)_2SO_4$ / $NH_4HSO_4$ and $NH_4NO_3$. As $NO_3^-$ had little change except in region 2

during cold season while $SO_4^{2-}$ decreased significantly during warm and cold seasons, SNA during cold season was getting comparable or even higher than SNA during warm season. Pollution of SNA in regions 2 and 3 was more of an issue during cold season than during warm season when $SO_2$ was further reduced during 2006-2016.

## 6. Summary and conclusion

With the implementation of the Title V of 1990 Amendments to the Clean Air Act of US in 1990s, emissions of $SO_2$ and $NO_x$ in the US reduced from 23.1 million tons/year to 3.7 million tons/year for $SO_2$ and from 25.2 million tons/year to 11.5 million tons/year for $NO_x$ from 1990 to 2015. In Canada, comparing to the emission level in 1990, $SO_2$ and $NO_x$ emissions in 2014 were reduced by 63% and

33% respectively, through the 1985 Eastern Canada Acid Rain Program, the 1991 Canada-United States Air Quality Agreement, and various federal and provincial/territorial regulations set up to reduce $SO_2$ and $NO_x$ emissions. In both the US and Canada, the reduction of the emissions was mainly in eastern regions of the countries. With the significant reduction of $SO_2$ and $NO_x$ emissions, air concentration of gases $SO_2$ and $HNO_3$, and particles $SO_4^{2-}$, $NO_3^-$ and $NH_4^+$ had a very different nonlinear response to the

emission reductions, both spatially and temporally. In this study, we analyzed the air concentration of $SO_4^{2-}$, $NO_3^-$, $NH_4^+$, $HNO_3$ and $SO_2$ measured weekly by the CASTNET in US and daily by the CAPMoN in Canada from 1989 to 2015 to reveal the temporal and spatial changes during the period. Four distinct regions with characteristic pattern of air quality in the eastern US and Eastern Canada were identified: the northern Eastern America (region 1), the Midwest (region 2), the Mid-Atlantic (region 3), and the

Southeastern US (region 4).



In the first 10-year period of 1990-2000, $SO_2$ and $SO_4^{2-}$ decreased by more than 20% except $SO_4^{2-}$ in region 4 during cold season and $SO_2$ in region 4 during warm season. $NH_4^+$ declined by 12% to 29.8% during both seasons except that region 4 during cold season had a negligible change. $NO_3^-$ increased in regions 1-4 during cold season, and changed very little during warm season. $HNO_3$ reduced in regions

1-3 by more than 9% during both seasons, and increased in region 4 by 5% and 5.7% during cold and warm seasons. In a 25-year period of 1990-2015, reduction of $SO_4^{2-}$ was from 60.1% in region 2 to 62.5% in region 3 during cold season, and from 72.7% in region 1 to 78.7% in region 4 during warm season. Reduction of $SO_2$ was the most significant among all species, ranging from 83.9% during warm season in region 4 to 91.2% during warm season for region 3. During warm season, reduction of $NO_3^-$

was seen in all regions, ranging from 14.3% in region 1 to 57.5% in region 2. During cold season, reduction of $NO_3^-$ was only observed in region 2 at 30.4%, and change of $NO_3^-$ was negligible in regions 3 and 4. Reduction of $HNO_3$ during warm season was relatively uniform in terms of percentage, ranging from 63.1% to 68.8%; during cold season, region 2 had the lowest percentage of reduction at 56.0%, and region 1 had the highest at 63.5%. Reduction of $NH_4^+$ was most significant during warm

season in terms of both percentage and absolute value. The reduction was ranging from 74.0% to 75.7% in regions 2-4 during warm season, more than 20% more than the reduction rate during cold season, which was in range of 48.9% to 53.2%. The time series of annual concentration during warm and cold seasons show that the reduction of each species was not even during the period. Reduction of $SO_4^{2-}$ and $SO_2$ mainly happened during 1989-1995 and 2007-2016 for warm season, and during

1989-1995 and 2005-2016 for cold season. Reduction of $NO_3^-$ was mainly after year 2000.

$RSO_4$ is a metric indicating the gas-aerosol partition of sulfur in the air, and $RNO_3$ is indicator of the percentage $HNO_3$ is aerosolized. $RSO_4$ increased by 48.6% to 166.3% during the period for cold season, and by 0.6% to 58.4% during warm season. $RSO_4$ was found to increase quadratically with decreasing of $SO_2$ for regions 2-4 and the two seasons. The significant increase of $RSO_4$ during cold season

explains why reduction rate of $SO_2$ was much higher than that of $SO_4^{2-}$ during the period, as reduction



of $SO_2$ was due to not only the emission reduction, but also more $SO_2$ was converted to $SO_4^{2-}$. Faster reduction of $SO_2$ was observed for all 4 regions during cold season, both before and after year 2000, and it was more obvious after 2000. During warm season, difference of reduction rate of $SO_2$ vs. $SO_4^{2-}$ was much less. This is because the increase of $RSO_4$ during warm season was much less significant than

5 during cold season. In regions 1 and 4, $RSO_4$ was only increased by 0.6% and 12.4% percentage during warm season. Differences of reduction rate of $SO_2$ vs. $SO_4^{2-}$ for these two regions during warm season were only 1.7% and 5.2% respectively. For regions 1, 3 and 4, $RNO_3$ increased between 79.2% and 94.9% during warm season, and between 73.9% and 92.3% during cold season. For region 2, $RNO_3$ for warm and cold seasons were only increased by 8.0% and 12.8% respectively as $NH_3$ in the region was

10 excess to neutralize $HNO_3$. $RNO_3$ was found to increase with decreasing of $SO_4^{2-}$ quadratically in regions 3 and 4.

In summary, with the significant reduction of $SO_2$ and $NO_x$ in EUS and EC during 1990-2016, $SO_4^{2-}$, $SO_2$, $NH_4^+$, and $HNO_3$ were reduced significantly by 73.3%, 87.6%, 67.4% and 65.8% for the whole region. Reduction of $NO_3^-$ was relatively less significant at 29.1%, and it mainly happened: (1) after year 2000;

(2) in regions 1-4 during warm season; (3) during cold season in region 2 only.





## Author contributions

JF carried out the overall analysis and interpretation of the data as well as wrote the manuscript. EC did the initial data and trend analysis. RV provided the supervision of the study and discussed the results.

## Competing interests

The authors declare that they have no conflict of interest.



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







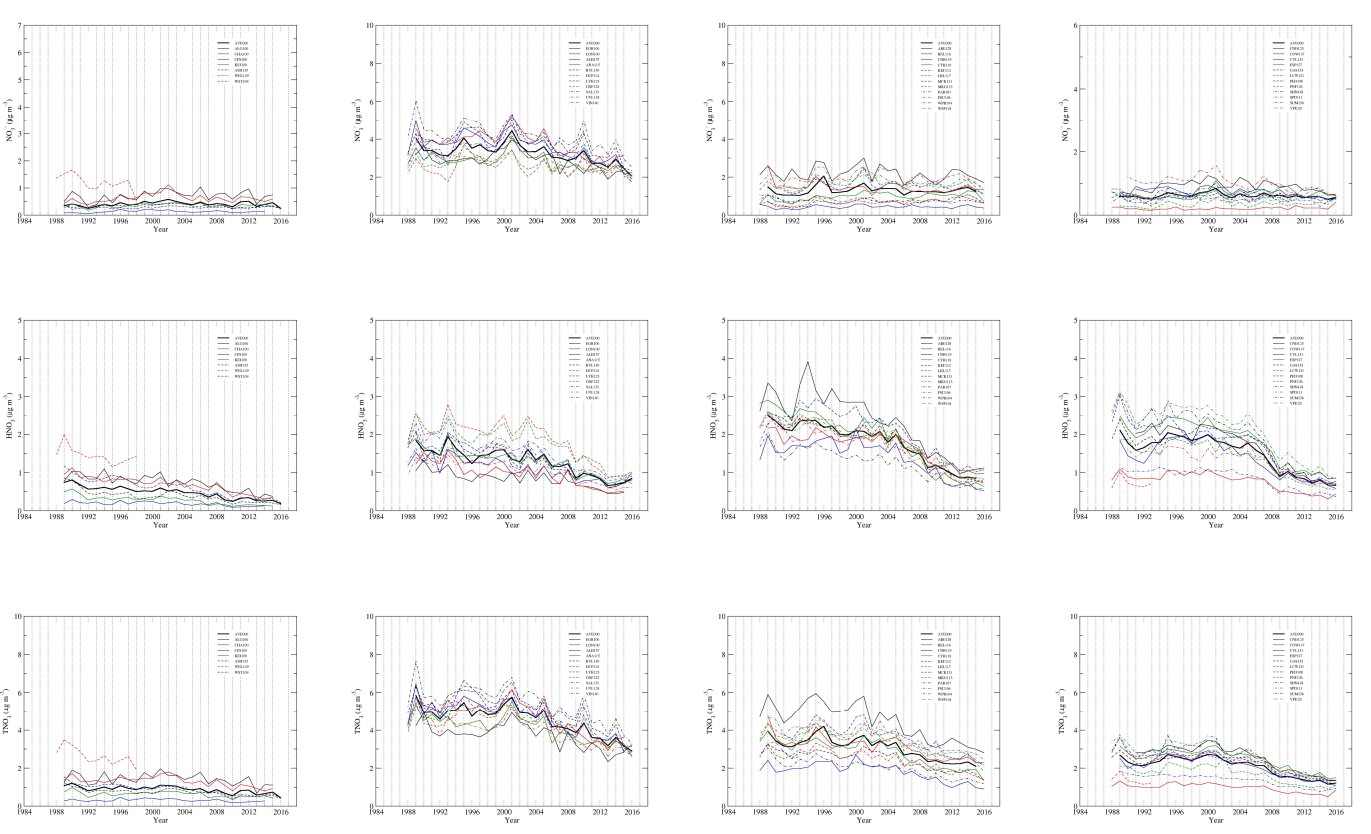

(a) Cold Season



Region 1          Region 2          Region 3          Region 4

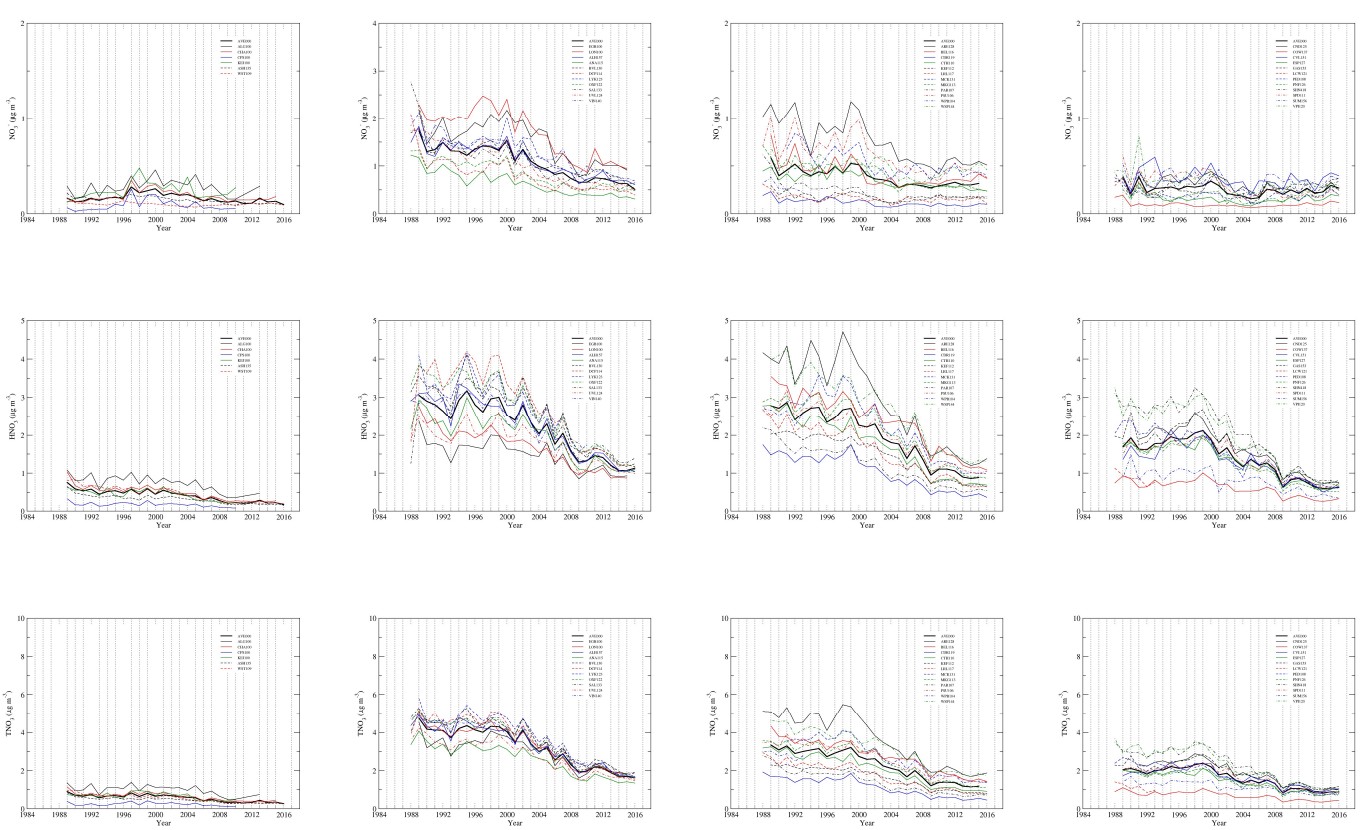

(b)  Warm season

Fig. 2 Time series of annual concentration during cold (a) and warm (b) seasons for each species and each region.



Region 1

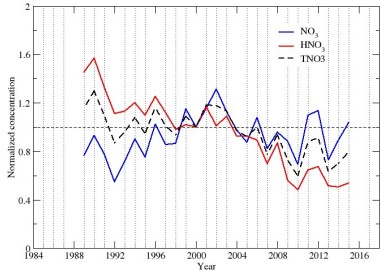
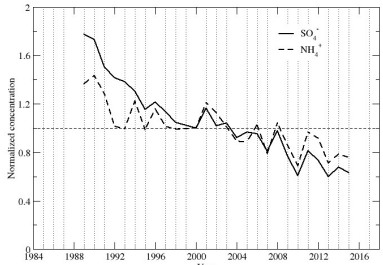
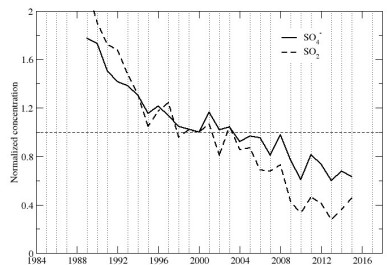

Region 2

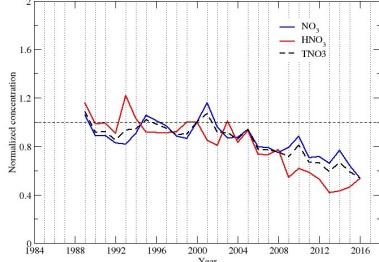
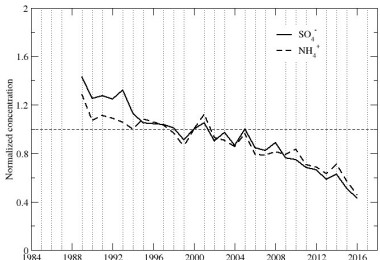
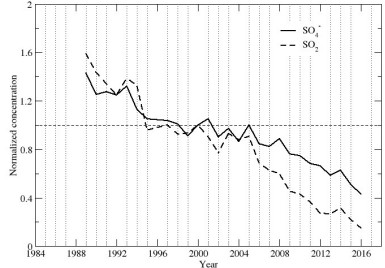





Region 3

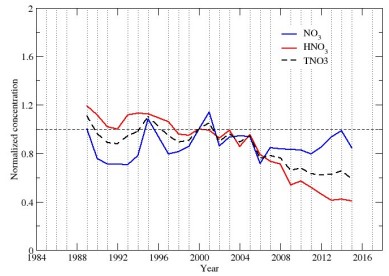

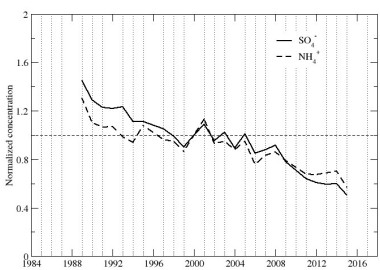

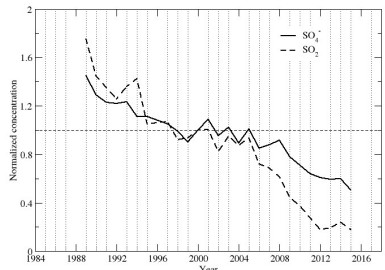

Region 4

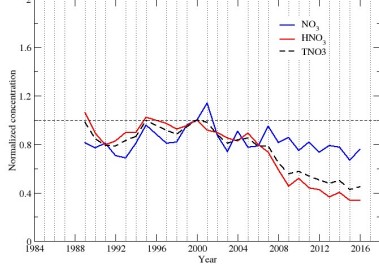

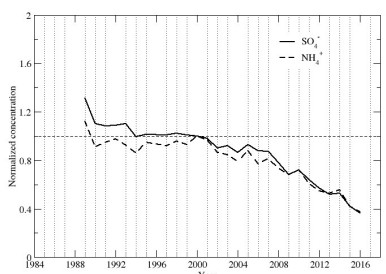

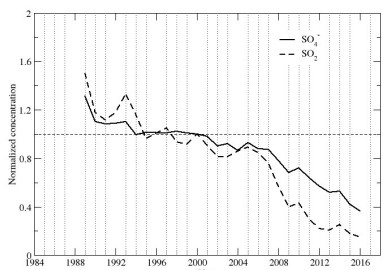

(a) Cold Season





Region 1

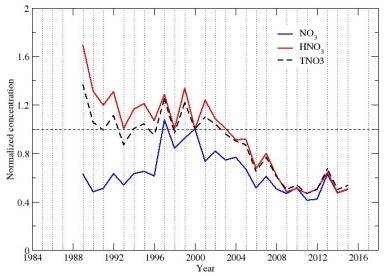
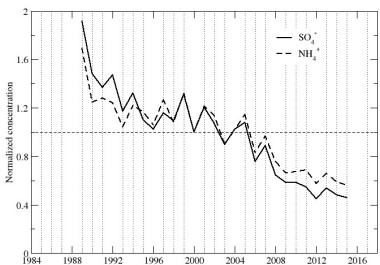
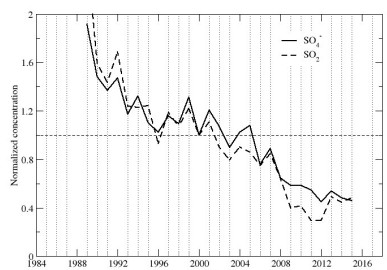

Region 2

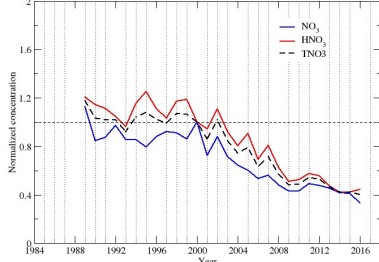
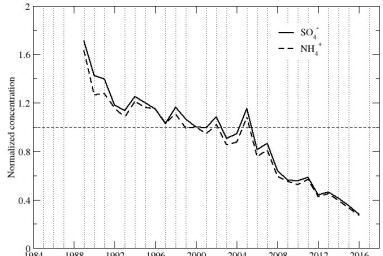
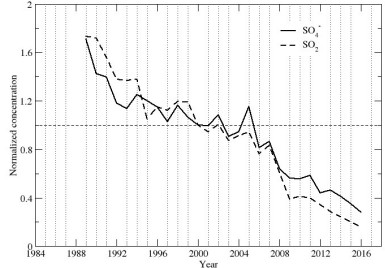





Region 3

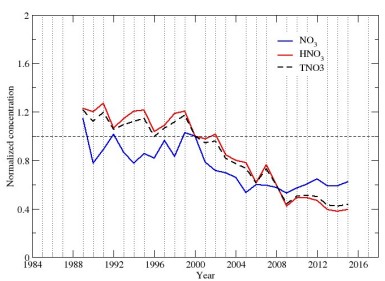 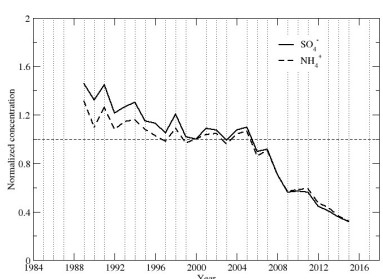 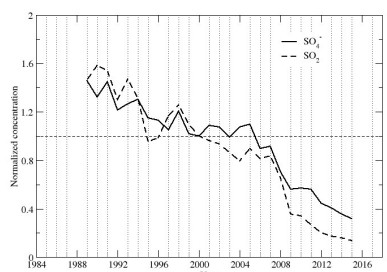

Region 4

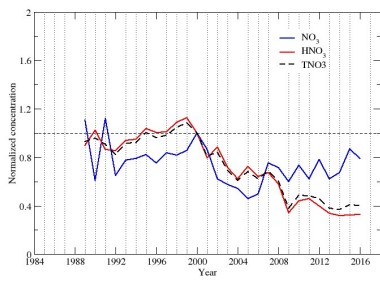 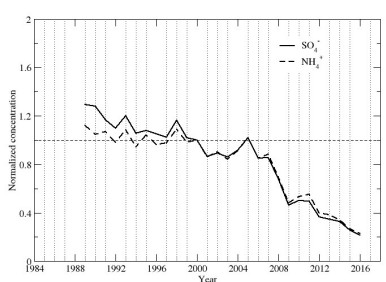 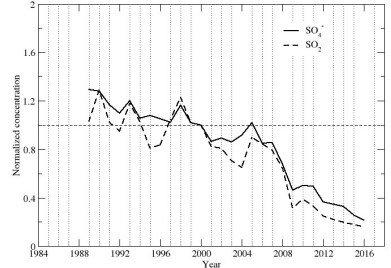

(b) Warm season

Fig. 3 Time series of regional annual concentration normalized to year 2000 for each species during cold (a) and warm (b) seasons.



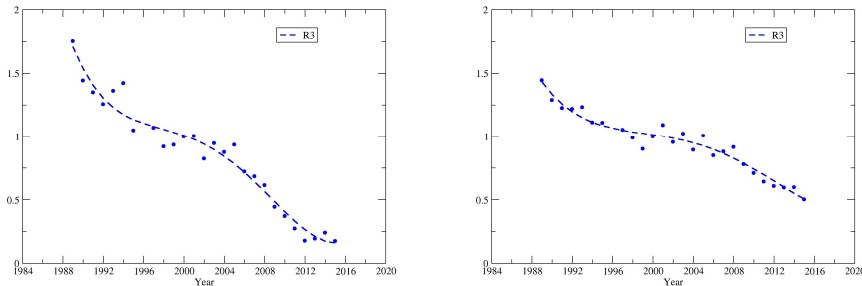

Fig. 4 Examples of $4^{th}$ Polynomial regressions of $SO_4^{2-}$ and $SO_2$ for region 3 during cold season.





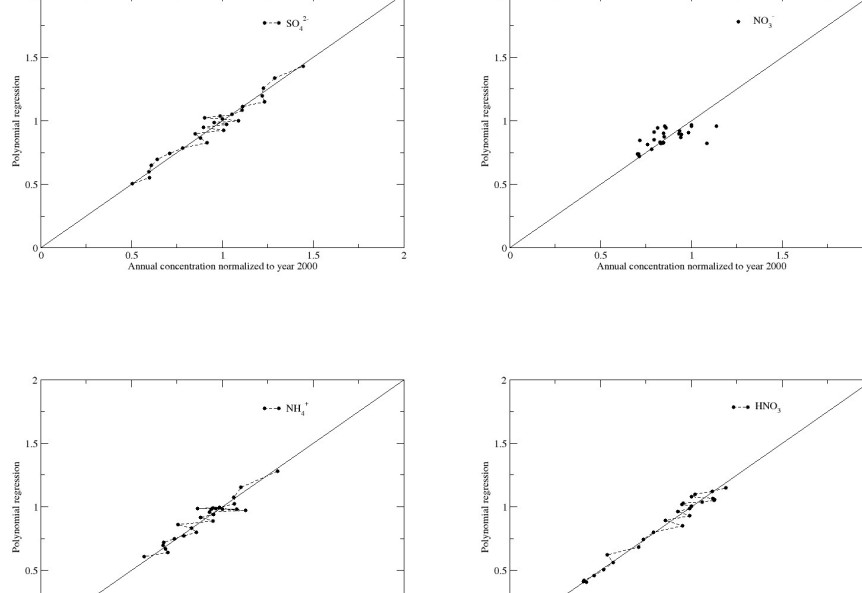





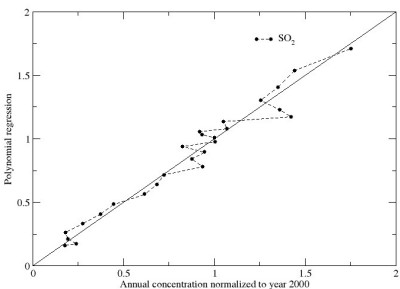

Fig. 5 Comparisons of normalized annual concentrations with the ones regressed with polynomial regressions for $SO_4^{2-}$, $SO_2$, $NH_4^+$, $NO_3^-$ and $HNO_3$ for region 3 during cold season. The dot lines link the annual concentrations from 1990 to 2016.



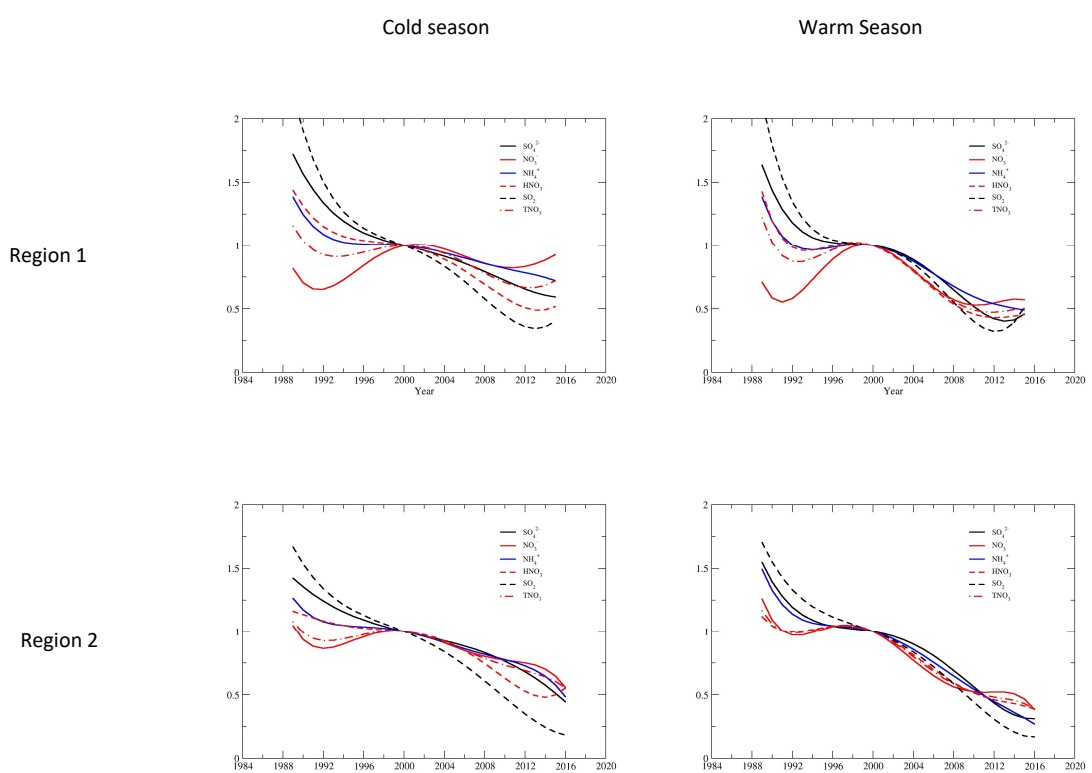





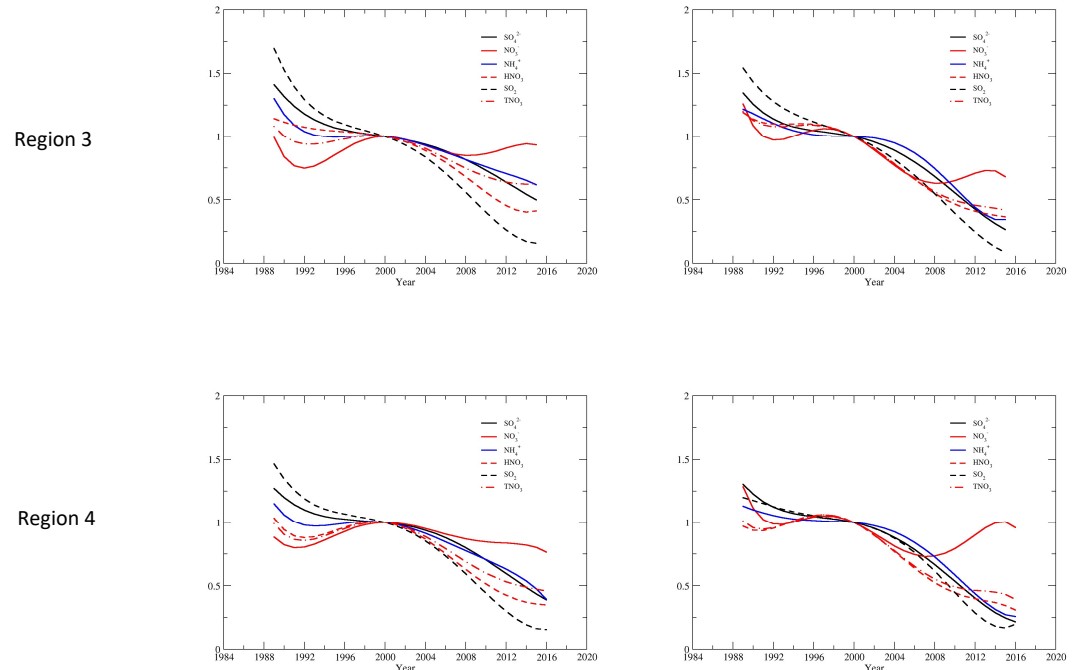

(a) Long-term trends for different species in each region



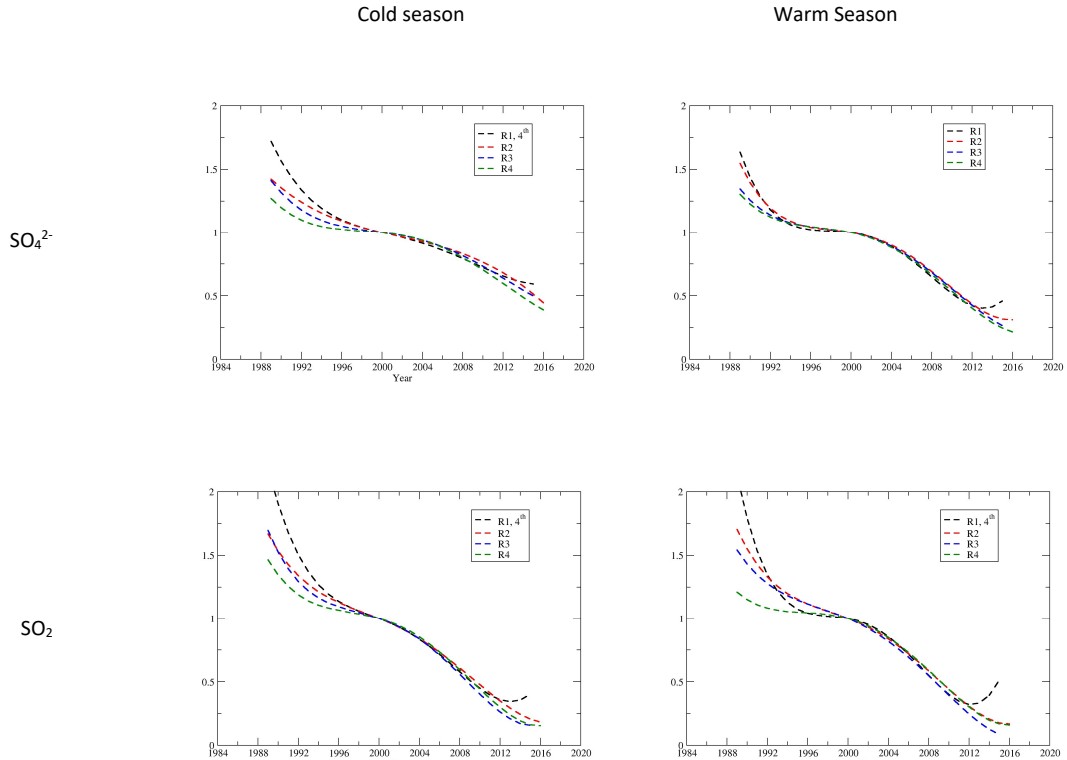





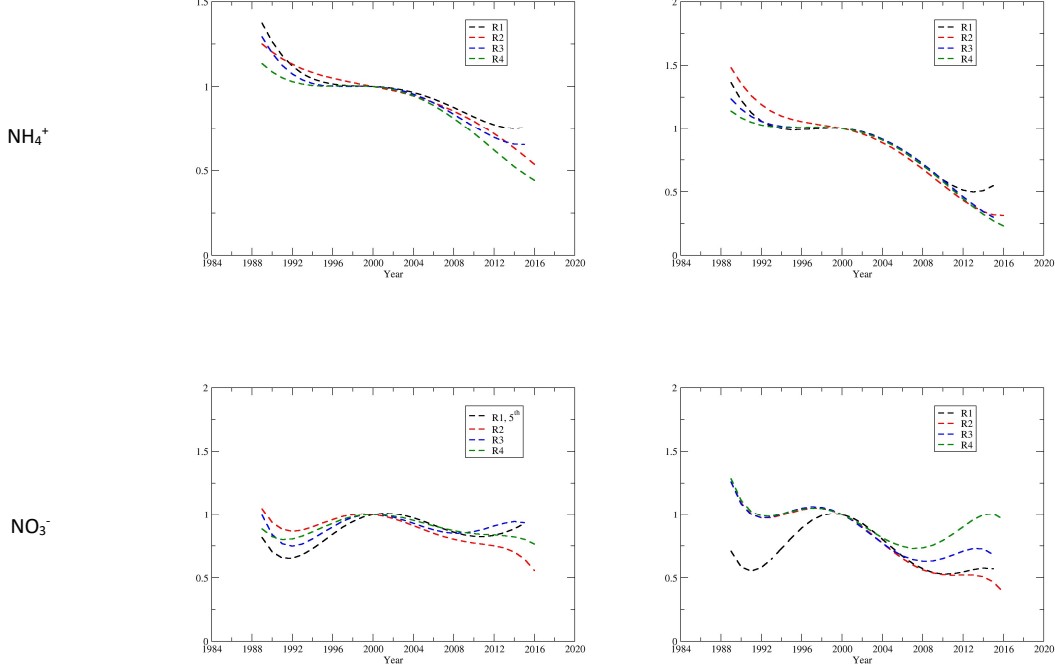



HNO₃

TNO₃

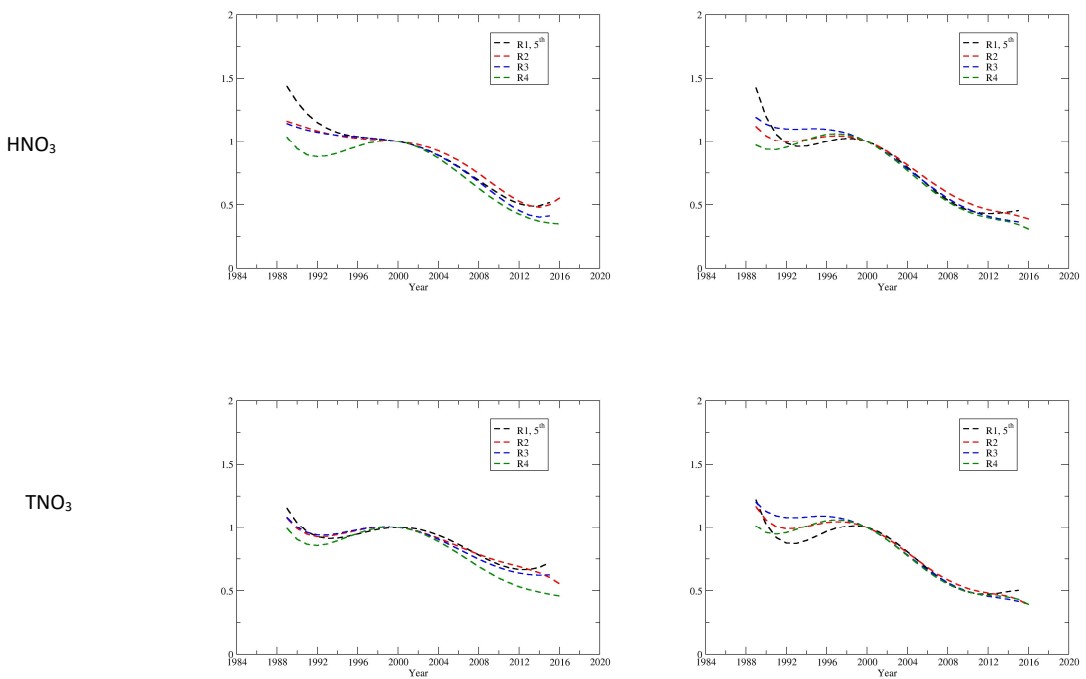

(b) Long-term trends for different region and each species

Fig. 6 Long-term trends derived with 4th polynomial regressions for $SO_4^{2-}$, $SO_2$ and $NH_4^+$, and 5th polynomial regressions for $NO_3^-$, $HNO_3$ and TNO3 during warm and cold seasons.

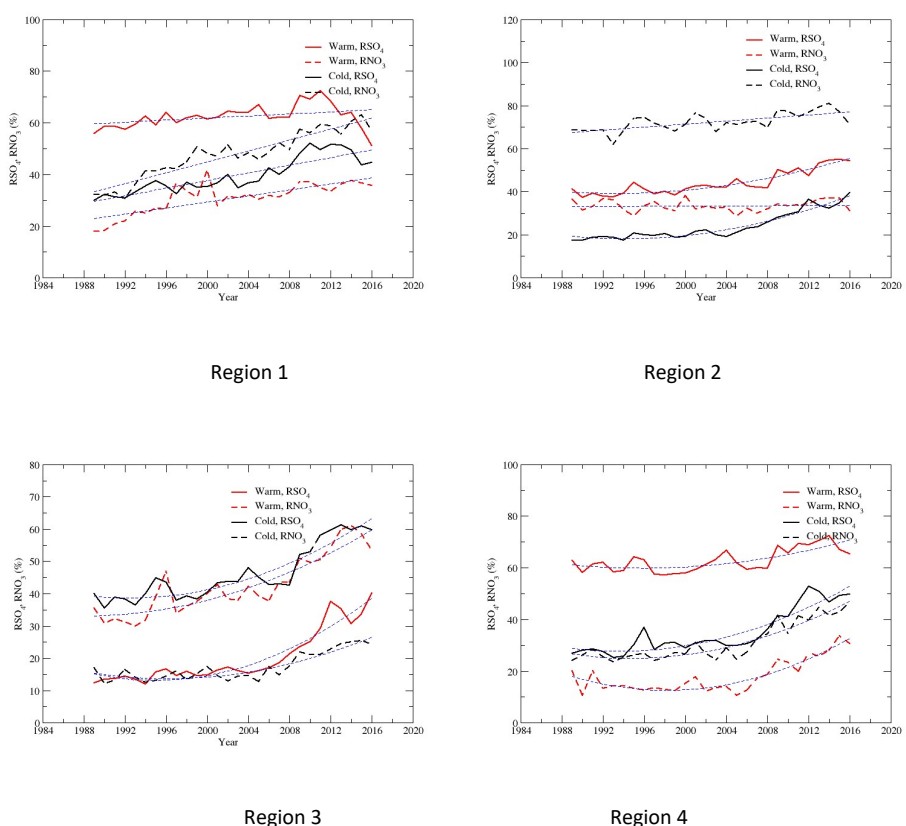

Fig. 7 Time series of $RSO_4$ and $RNO_3$ for regions 1-4 during cold and warm seasons.





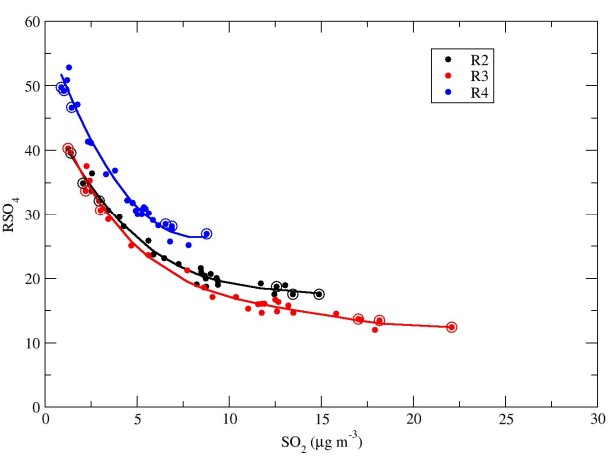

(a) Cold season

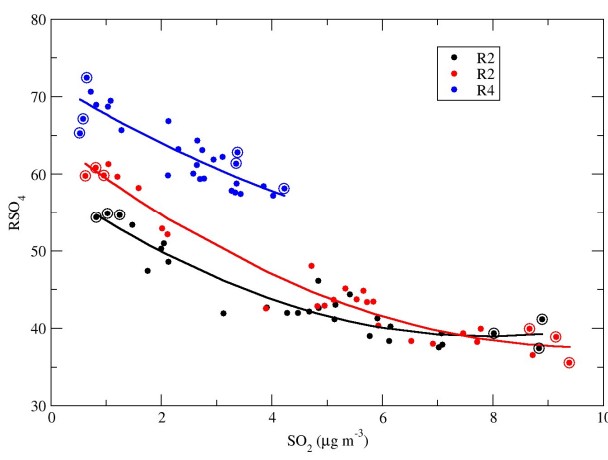

(b) Warm season

Fig. 8 Correlations of annual mean during cold and warm seasons: RSO₄ vs. SO₂ for regions 2-4.





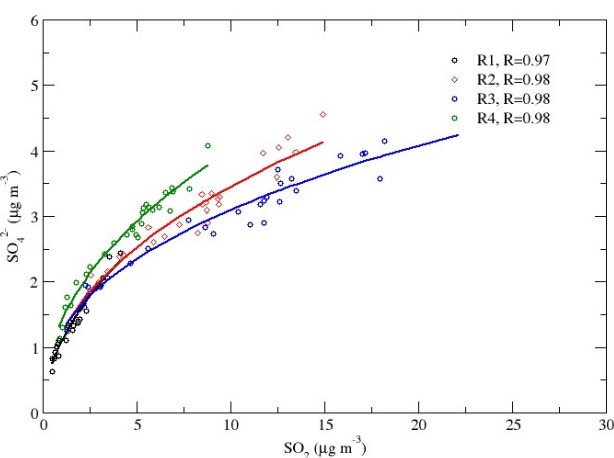

(a)  Cold season

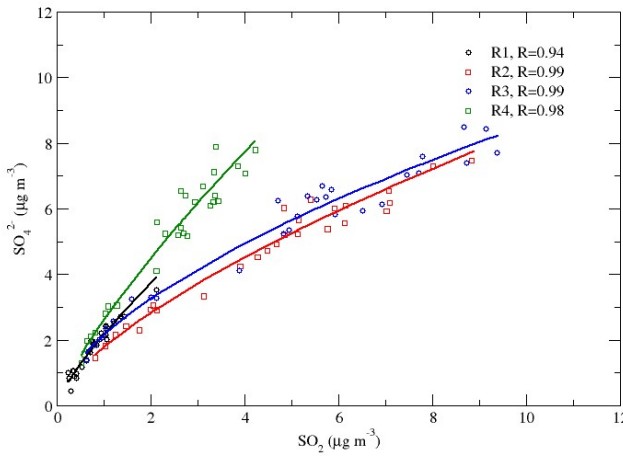

(b)  Warm season

Fig. 9 Correlations of annual mean during cold and warm seasons: SO$_4^{2-}$ vs. SO$_2$ for regions 1-4.





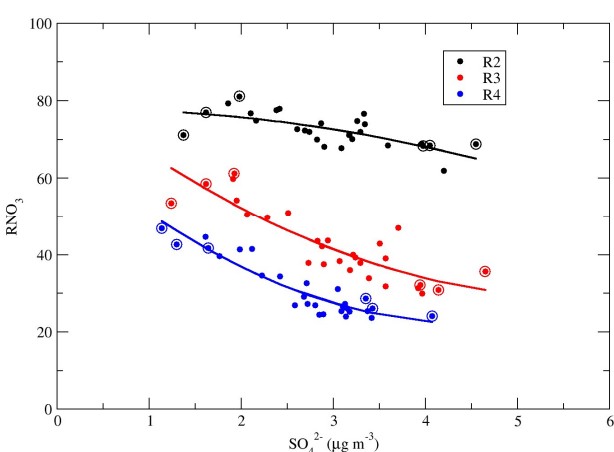

(a) Cold season

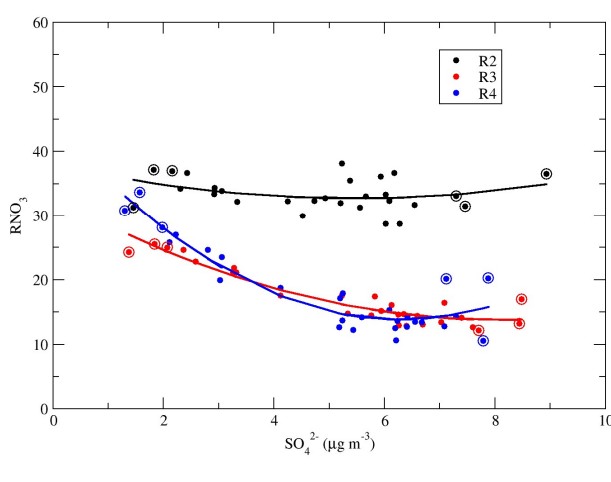

(b) Warm season

Fig. 10 Correlations of annual mean during cold and warm seasons: RNO$_3$ vs. SO$_4^{2-}$ for regions 2-4.




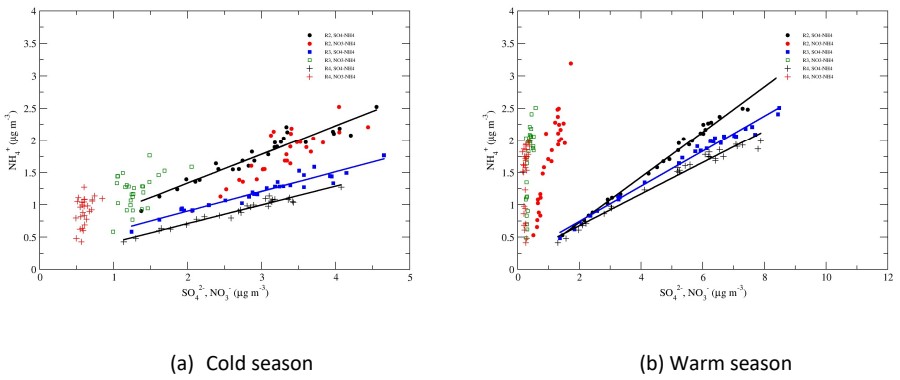

(a)  Cold season                                    (b) Warm season

Fig. 11 Correlations of $NH_4^+$ vs. $SO_4^{2-}$ and $NH_4^+$ vs. $NO_3^-$ during cold and warm seasons



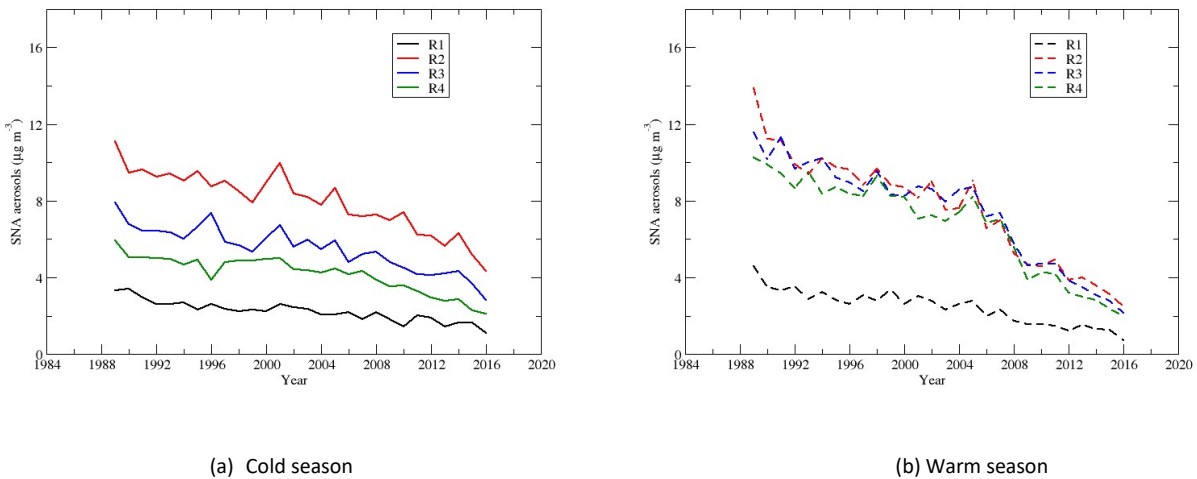

(a) Cold season                                (b) Warm season

Fig. 12 Time series of annual mean of sulfate-nitrate-ammonium aerosols during cold and warm seasons.





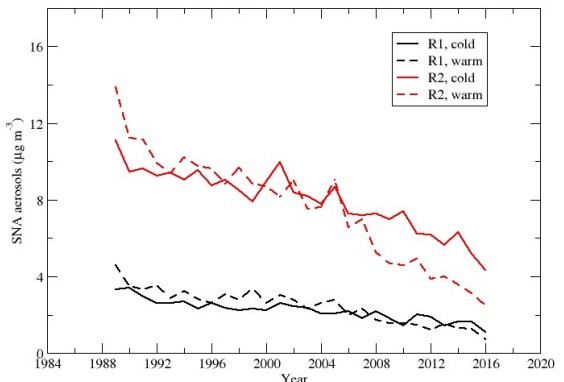
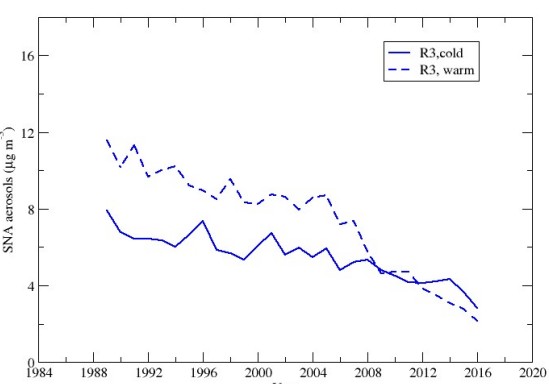



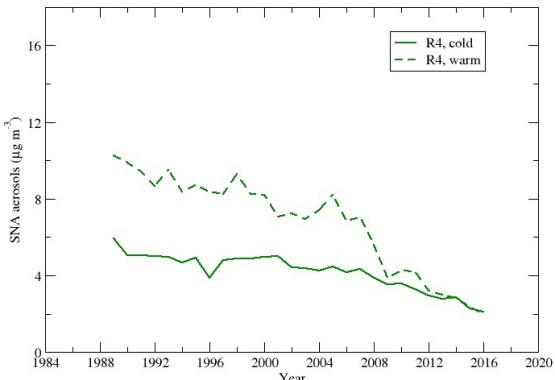

Fig. 13 Time series of annual mean of sulfate-nitrate-ammonium aerosols in regions 1-4.



Table 1. Characteristics of 4 regions based on 3-year averages of 1989-1991 during cold season.

| Region | Characteristics of region |
|---|---|
| 1 | $SO_2 < 6.4$ µg µ$^{-3}$ for all sites; in average the region had lowest annual concentrations of $SO_4^{2-}$, $NO_3^-$, $NH_4^+$, $HNO_3$, and $SO_2$. |
| 2 | $NO_3^- > 2.5$ µg m$^{-3}$ and $RNO_3 > 54.0\%$ for all sites; $AVE\_NO_3^- = 4.2$ µg m$^{-3}$; $AVE\_SO_2 = 13.6$ µg m$^{-3}$; $AVE\_RNO_3 = 68.5\%$. |
| 3 | $NO_3^- < 2.2$ µg m$^{-3}$, $RNO_3 < 47\%$ and $SO_2 > 15.2$ µg m$^{-3}$ for all sites; $AVE\_NO_3^- = 1.2$ µg m$^{-3}$; $AVE\_SO_2 = 19.2$ µg m$^{-3}$; $AVE\_RNO_3 = 32.3\%$. |
| 4 | $SO_2 < 11.7$ µg m$^{-3}$ and $NO_3^- < 0.7$ µg m$^{-3}$ for all sites; $AVE\_NO_3^- = 0.6$ µg m$^{-3}$; $AVE\_SO_2 = 7.2$ µg m$^{-3}$; $AVE\_RNO_3 = 28.3\%$. |



Table 2. Changes of air concentration of pollutants (µg m$^{-3}$), RSO$_4$ (%) and RNO$_3$ (%) between 1989-1991 and 1999-2001 for regions 1-4.

(a) Cold season

| Region | | SO$_4^{2-}$ | NO$_3^-$ | NH$_4^+$ | HNO$_3$ | SO$_2$ | TNO$_3$ | RSO$_4$ | RNO$_3$ |
|---|---|---|---|---|---|---|---|---|---|
| 1 | 1989 - 1991 | 2.29 | 0.37 | 0.57 | 0.74 | 3.62 | 1.1 | 31.13 | 32.53 |
| | 1999 - 2001 | 1.46 | 0.49 | 0.44 | 0.54 | 1.91 | 1.01 | 35.7 | 48.58 |
| | Δ | -0.83 | 0.12 | -0.13 | -0.2 | -1.71 | -0.09 | 4.57 | 16.05 |
| | Δ % | **-36.2%** | **32.4%** | **-22.8%** | **-27.0%** | **-47.2%** | **-8.2%** | **14.7%** | **49.3%** |
| 2 | 1989 - 1991 | 4.19 | 3.62 | 2.27 | 1.66 | 13.65 | 5.25 | 17.92 | 68.49 |
| | 1999 - 2001 | 3.14 | 3.86 | 1.95 | 1.51 | 8.87 | 5.35 | 19.77 | 71.93 |
| | Δ | -1.05 | 0.24 | -0.32 | -0.15 | -4.78 | 0.1 | 1.85 | 3.44 |
| | Δ % | **-25.1%** | **6.6%** | **-14.1%** | **-9.0%** | **-35.0%** | **1.9%** | **10.3%** | **5.0%** |
| 3 | 1989 - 1991 | 4.24 | 1.21 | 1.56 | 2.31 | 19.2 | 3.49 | 13.11 | 32.76 |
| | 1999 - 2001 | 3.21 | 1.48 | 1.35 | 2.05 | 12.35 | 3.5 | 15.27 | 40.13 |
| | Δ | -1.03 | 0.27 | -0.21 | -0.26 | -6.85 | 0.01 | 2.16 | 7.37 |
| | Δ % | **-24.3%** | **22.3%** | **-13.5%** | **-11.3%** | **-35.7%** | **0.3%** | **16.5%** | **22.5%** |
| 4 | 1989 - 1991 | 3.57 | 0.55 | 1.09 | 1.81 | 7.24 | 2.33 | 28.29 | 25.14 |
| | 1999 - 2001 | 3.09 | 0.77 | 1.1 | 1.9 | 5.52 | 2.66 | 30.36 | 28.31 |
| | Δ | -0.48 | 0.22 | 0.01 | 0.09 | -1.72 | 0.33 | 2.07 | 3.17 |
| | Δ % | **-13.4%** | **40.0%** | **0.9%** | **5.0%** | **-23.8%** | **14.2%** | **7.3%** | **12.6%** |



(b) Warm season

| Region | | $SO_4^{2-}$ | $NO_3^-$ | $NH_4^+$ | $HNO_3$ | $SO_2$ | $TNO_3$ | $RSO_4$ | $RNO_3$ |
|---|---|---|---|---|---|---|---|---|---|
| 1 | 1989 - 1991 | 2.93 | 0.14 | 0.73 | 0.62 | 1.56 | 0.75 | 57.71 | 19 |
| | 1999 - 2001 | 2.16 | 0.23 | 0.62 | 0.53 | 0.94 | 0.73 | 62.18 | 33.52 |
| | Δ | -0.77 | 0.09 | -0.11 | -0.09 | -0.62 | -0.02 | 4.47 | 14.52 |
| | Δ % | **-26.3%** | **64.3%** | **-15.1%** | **-14.5%** | **-39.7%** | **-2.7%** | **7.7%** | **76.4%** |
| 2 | 1989 - 1991 | 7.9 | 1.46 | 2.72 | 2.93 | 8.58 | 4.34 | 39.3 | 33.68 |
| | 1999 - 2001 | 5.33 | 1.32 | 1.91 | 2.65 | 5.37 | 3.93 | 40.73 | 33.76 |
| | Δ | -2.57 | -0.14 | -0.81 | -0.28 | -3.21 | -0.41 | 1.43 | 0.08 |
| | Δ % | **-32.5%** | **-9.6%** | **-29.8%** | **-9.6%** | **-37.4%** | **-9.4%** | **3.6%** | **0.2%** |
| 3 | 1989 - 1991 | 8.2 | 0.47 | 2.33 | 2.79 | 9.11 | 3.22 | 37.98 | 13.91 |
| | 1999 - 2001 | 6.04 | 0.48 | 1.91 | 2.4 | 6.06 | 2.84 | 40.7 | 15.74 |
| | Δ | -2.16 | 0.01 | -0.42 | -0.39 | -3.05 | -0.38 | 2.72 | 1.83 |
| | Δ % | **-26.3%** | **2.1%** | **-18.0%** | **-14.0%** | **-33.5%** | **-11.8%** | **7.2%** | **13.2%** |
| 4 | 1989 - 1991 | 7.61 | 0.32 | 1.92 | 1.74 | 3.67 | 2.04 | 60.76 | 16.99 |
| | 1999 - 2001 | 5.85 | 0.31 | 1.69 | 1.84 | 3.1 | 2.12 | 58.29 | 15.22 |
| | Δ | -1.76 | -0.01 | -0.23 | 0.1 | -0.57 | 0.08 | -2.47 | -1.77 |
| | Δ % | **-23.1%** | **-3.1%** | **-12.0%** | **5.7%** | **-15.5%** | **3.9%** | **-4.1%** | **-10.4%** |





Table 3. Changes of air concentration of air pollutants (μg m$^{-3}$), RSO$_4$ (%) and RNO$_3$ (%) between 1989-1991 and 2014-2016 for the eastern US and Eastern Canada. Red color in (a) indicates the 3-year average concentrations still exceeding 1.0 μg m$^{-3}$ during 2014-2016. Red color in (b) and (c) indicates that the reduction/increase rates exceeding 50%.

(a) All regions in the eastern US and Eastern Canada

| | | $SO_4^{2-}$ | $NO_3^-$ | $NH_4^+$ | $HNO_3$ | $SO_2$ | $TNO_3$ | $RSO_4$ | $RNO_3$ |
|---|---|---|---|---|---|---|---|---|---|
| All seasons | 1989 - 1991 | 5.44 | 1.1 | 1.78 | 1.96 | 8.73 | 3.02 | 33.6 | 31.72 |
| | 2014 - 2016 | 1.45 | 0.78 | 0.58 | 0.67 | 1.08 | 1.43 | 50.6 | 51.91 |
| | Δ | -3.99 | -0.32 | -1.2 | -1.29 | -7.65 | -1.59 | 17 | 20.19 |
| | Δ % | **-73.3%** | **-29.1%** | **-67.4%** | **-65.8%** | **-87.6%** | **-52.6%** | **50.6%** | **63.7%** |
| Cold season | 1989 - 1991 | 3.73 | 1.58 | 1.49 | 1.76 | 11.71 | 3.31 | 21.41 | 40.98 |
| | 2014 - 2016 | 1.4 | 1.21 | 0.68 | 0.66 | 1.5 | 1.86 | 42.33 | 61.04 |
| | Δ | -2.33 | -0.37 | -0.81 | -1.1 | -10.21 | -1.45 | 20.92 | 20.06 |
| | Δ % | **-62.5%** | **-23.4%** | **-54.4%** | **-62.5%** | **-87.2%** | **-43.8%** | **97.7%** | **49.0%** |
| Warm season | 1989 - 1991 | 7.02 | 0.66 | 2.06 | 2.18 | 6.15 | 2.81 | 47.86 | 21.29 |
| | 2014 - 2016 | 1.6 | 0.41 | 0.53 | 0.75 | 0.75 | 1.15 | 60.4 | 34.32 |
| | Δ | -5.42 | -0.25 | -1.53 | -1.43 | -5.4 | -1.66 | 12.54 | 13.03 |
| | Δ % | **-77.2%** | **-37.9%** | **-74.3%** | **-65.6%** | **-87.8%** | **-59.1%** | **26.2%** | **61.2%** |





(b)  Regions 1-4 during warm season

| Region | | SO$_4^{2-}$ | NO$_3^-$ | NH$_4^+$ | HNO$_3$ | SO$_2$ | TNO$_3$ | RSO$_4$ | RNO$_3$ |
|---|---|---|---|---|---|---|---|---|---|
| 1 | 1989 - 1991 | 2.93 | 0.14 | 0.73 | 0.62 | 1.56 | 0.75 | 57.71 | 19.0 |
| | 2014 - 2016 | 0.8 | 0.12 | 0.28 | 0.21 | 0.4 | 0.33 | 58.03 | 37.04 |
| | Δ | -2.13 | -0.02 | -0.45 | -0.41 | -1.16 | -0.42 | 0.32 | 18.04 |
| | Δ % | **-72.7%** | -14.3% | **-61.6%** | **-66.1%** | **-74.4%** | **-56.0%** | 0.6% | **94.9%** |
| 2 | 1989 - 1991 | 7.9 | 1.46 | 2.72 | 2.93 | 8.58 | 4.34 | 39.3 | 33.68 |
| | 2014 - 2016 | 1.83 | 0.62 | 0.66 | 1.08 | 1.03 | 1.68 | 54.85 | 36.38 |
| | Δ | -6.07 | -0.84 | -2.06 | -1.85 | -7.55 | -2.66 | 15.55 | 2.7 |
| | Δ % | **-76.8%** | **-57.5%** | **-75.7%** | **-63.1%** | **-88.0%** | **-61.3%** | 39.6% | 8.0% |
| 3 | 1989 - 1991 | 8.2 | 0.47 | 2.33 | 2.79 | 9.11 | 3.22 | 37.98 | 13.91 |
| | 2014 - 2016 | 1.77 | 0.3 | 0.6 | 0.87 | 0.8 | 1.16 | 60.15 | 24.92 |
| | Δ | -6.43 | -0.17 | -1.73 | -1.92 | -8.31 | -2.06 | 22.17 | 11.01 |
| | Δ % | **-78.4%** | -36.2% | **-74.2%** | **-68.8%** | **-91.2%** | **-64.0%** | **58.4%** | **79.2%** |
| 4 | 1989 - 1991 | 7.61 | 0.32 | 1.92 | 1.74 | 3.67 | 2.04 | 60.76 | 16.99 |
| | 2014 - 2016 | 1.62 | 0.27 | 0.5 | 0.61 | 0.59 | 0.86 | 68.29 | 30.8 |
| | Δ | -5.99 | -0.05 | -1.42 | -1.13 | -3.08 | -1.18 | 7.53 | 13.81 |
| | Δ % | **-78.7%** | -15.6% | **-74.0%** | **-64.9%** | **-83.9%** | **-57.8%** | 12.4% | **81.3%** |



(c)  Regions 1-4 during cold season

| Region | | $SO_4^{2-}$ | $NO_3^-$ | $NH_4^+$ | $HNO_3$ | $SO_2$ | $TNO_3$ | $RSO_4$ | $RNO_3$ |
|---|---|---|---|---|---|---|---|---|---|
| 1 | 1989 - 1991 | 2.29 | 0.37 | 0.57 | 0.74 | 3.62 | 1.1 | 31.13 | 32.53 |
| | 2014 - 2016 | 0.87 | 0.44 | 0.32 | 0.27 | 0.75 | 0.71 | 46.25 | 62.54 |
| | Δ | -1.42 | 0.07 | -0.25 | -0.47 | -2.87 | -0.39 | 15.12 | 30.01 |
| | Δ % | **-62.0%** | 18.9% | -43.9% | **-63.5%** | **-79.3%** | -35.5% | **48.6%** | **92.3%** |
| 2 | 1989 - 1991 | 4.19 | 3.62 | 2.27 | 1.66 | 13.65 | 5.25 | 17.92 | 68.49 |
| | 2014 - 2016 | 1.67 | 2.52 | 1.16 | 0.73 | 2.19 | 3.24 | 35.18 | 77.29 |
| | Δ | -2.52 | -1.1 | -1.11 | -0.93 | -11.46 | -2.01 | 17.26 | 8.8 |
| | Δ % | **-60.1%** | -30.4% | -48.9% | **-56.0%** | **-84.0%** | -38.3% | **96.3%** | 12.8% |
| 3 | 1989 - 1991 | 4.24 | 1.21 | 1.56 | 2.31 | 19.2 | 3.49 | 13.11 | 32.76 |
| | 2014 - 2016 | 1.59 | 1.25 | 0.77 | 0.86 | 2.17 | 2.09 | 34.91 | 57.95 |
| | Δ | -2.65 | 0.04 | -0.79 | -1.45 | -17.03 | -1.4 | 21.8 | 25.19 |
| | Δ % | **-62.5%** | 3.3% | **-50.6%** | **-62.8%** | **-88.7%** | -40.1% | **166.3%** | **76.9%** |
| 4 | 1989 - 1991 | 3.57 | 0.55 | 1.09 | 1.81 | 7.24 | 2.33 | 28.29 | 25.14 |
| | 2014 - 2016 | 1.36 | 0.55 | 0.51 | 0.71 | 1.13 | 1.25 | 48.53 | 43.71 |
| | Δ | -2.21 | 0 | -0.58 | -1.1 | -6.11 | -1.08 | 20.24 | 18.57 |
| | Δ % | **-61.9%** | 0.0% | **-53.2%** | **-60.8%** | **-84.4%** | -46.4% | **71.5%** | **73.9%** |