# Peer review of "Air quality in the eastern United States and Eastern Canada for 1990-2015: 25 years of change in response to emission reductions of SO2 and NOx in the region"

_Atmospheric Chemistry and Physics, 2019_

## Referee Comment (RC1) · Tom Butler (Referee) · 13 Sep 2019

I found the article by Feng et al. (Air quality in the eastern US and Eastern Canada for 1990-2015:.. ... scientifically appropriate and worth publication with some revisions. This paper addresses the changes in air quality that have occurred during a 25 year (or 27 year - there are data here that extend from 1990 to 2016) period in which emissions of NOx and SO2 have been substantially reduced in eastern North America. The paper's focus is on: 1) declines in air concentrations, of gaseous SO2 and HNO3, particulate SO4=, NO3- and NH4+ (all measured by the U.S. CASTNET and Canadian

[Figure]

CAPMoN networks) in four distinct regions, 2) the seasonal (cold versus warm season) differences, and 3) the temporal changes in partitioning of gaseous and particulate species. There are quite a few grammatical errors, and I have corrected some but not all of them. A careful reading of the text by the authors to correct this is recommended.

My comments for improvement of the manuscript are: 1) Pg 4 lines 10 to 15 . Break this up into two sentences.

2) Pg 4 line 20. The sharp increase in 2002 (not 2022) in NOx from on road vehicles is NOT due to a true increase in emissions. It is due to a change in methodology on how the emissions are calculated. The increase is a methodological change, and EPA does not go back and correct for that.

3) Pg 6 line 19. Environmental Protection Agency 4) Pg 7 line 20. . . ...is more stable. . ..

5) Pg 7 line 25. Figures S1 and S2 would be more useful to the reader if 89-91 data appear side by side with the 2014 –

6) 2016 data (as is done in Fig. S3).

7) Section 3.2 pg 10 lines 17 and 18. . . ..Fig. 2.a for the cold season and Fig. 2.b for the warm season. . .. (cold and warm are in the wrong order here)

8) Section 3.2 This section had too much year to year (or 3 year etc.) detail that can lose the reader. This section should be shortened, skipping minor changes that don't address the overall patterns. The 3-year periods are too short and confusing when you are dealing with a 25 year record. The general trends are what is of interest. For example, in Section 3.2.2.2 pg 15 lines 6 to 9 are more concise and give the overall pattern.

9) Pg 18 line 5. . . ..NH4+ was reduced by 12% and 29.8% during the cold and warm seasons, respectively. . .. (or is it the warm and cold season, respectively? It is not clear).

10) Pg 18 line 7. In general, for the first ten-year period, 1990-1999, SO2. . ... .

11) Pg 18 line 14. During the 25-year period of 1990-2015, air quality. . ...

12) Pg 19 lines 24 & 25. The reduction during the warm season was much greater than in the cold season, ranging from a reduction of 11.4% in region 4 to a reduction of 23.9% in region 3.

13) Pg 20 line 19 & 20. SO4= was reduced by 73.3% for the whole region.

14) Pg 22 line 14. decreased

15) Pg 22 line 19-20. . ...5th order polynomial. . ..

16) Pg 23 lines 8 and 9. . ...differences in trends between 1989-2000 for 1 and 4.

17) Pg 25 line 8. There is a disparity in the reduction of SO2 and SO4= concentrations in response to emission reductions of SO2, . ...

18) Pg 25 line21. . ...in the first ten years, 1990 to 2000, but it was. . ...

19) Pg 26 lines 10 and 11. The text, referring to Fig. 9, describes "linear regressions, yet the relations shown appear to be curvilinear (and at the end of the paragraph (pg 27) the correlation of S04= vs. SO2 is described as a power-law relationship. "Linear regression" appears to be incorrect.

20) Pg 27 line 8. . ...sea salts. . ..

21) Pg 29 lines 22. and 23 . ...the trend was reversed after that.

22) Pg 30 line 3. . ...decreasing trend even though the reduction. . ...

23) Pg 30 line 12. . ...SO2 and NOx in the US were reduced. . ..

24) Pg 32 lines 9 and 10. . ...in the region was in excess. . ...

25) Figure 4. Each graph should have a label for the Y-axis (SO4=, SO2) or have a label that makes clear what the dependent variable is.

26) Figure 6. caption ….derived from 4th order polynomial…. ….and 5th order polynomial…

27) Figure 8 (b). the legend has R2,R2,R4. It should be R2,R3,R4. Why are some of the data points circled in both (a) and (b)? Explain in caption.

28) Figure 9. Why are some of the data points circled in both (a) and (b)? Explain in caption.

29) Figure 11. Explain in caption why some sets of data points have regression lines, and others do not.

30) Figure 13. Can the 2nd and 3rd graphs be combined, as is the case with the first graph?

General comment - The authors have more significant figures than they need in many places throughout the text, especially when referring to percentages (eg. SO4= was reduced 73% rather than 73.3%). Correcting this throughout the text would be a good idea. Overall, this is a very good paper that demonstrates the dramatic change in air quality brought about by clean air legislation in the US and Canada. Some grammatical work would improve the clarity of the presentation.

---

## Referee Comment (RC2) · Anonymous Referee #1 · 24 Oct 2019

**A review on the manuscript ACP-2019-567 entitled "Air quality in the eastern United States and Eastern Canada for 1990-2015: 25 years of change in response to emission reductions of SO2 and NOx in the region".**

The paper is relevant for understanding the impact of emission reduction on air quality across eastern-US and Canada, between 1990 and 2015, providing a temporal and spatial analysis of observational data, and what are the possible chemical and physical mechanisms responsible for the that evolution.

General:

The study presented is very rich in data and analysis. However, some of the information could be summarized making the manuscript more readable, especially in the results section. Every sub-section of the results section has a summary after the description of the results, the summary is good but the text prior to it is overwhelming and unnecessary when having adequate figures/tables supporting the text. The figures and tables should be more carefully thought through, as it takes a lot of time to grasp the information.

The reader might have issues to grasp what was done every single step of the way to obtain the results and the analysis, as the methodology is not really clear. This might be a consequence of not having a dedicated section for the methodology. Additionally, it might be challenging the reading when interchanging time periods and not giving a clear context, especially when trends and changes in concentrations are calculated based on different time period, and not really explained why.

More information is needed to understand the levels of concentration behind the clusters, this might be due to emissions, meteorology, terrain, etc. It is important to know if one cluster can be more populated than others in terms of sources, and if that is still the case throughout the study period.

**Detailed:**

Section 1

P2/L20: the reference to WHO fact sheets should be substituted by papers available describing the impact of air pollution on human health.

P3/L4-P4/L25: the authors appropriately describe how emissions have been evolving throughout the study period and the legislation to enforce such changes, but there is no single reference to corroborate the numbers. Please add the references grounding your narrative. Additionally, only the US situation was described, there is similar discourse for Canada. Please revise.

P5/L1-4: the authors describe trend analysis studies for other regions than US, and this might be misleading, even unnecessary. Please consider to remove.

Chemistry Warm/cold

**Section 2.1**

P6/L25 Please add a reference describing the CASTNET network

The authors describe the temporal resolution of the measurements available for each network, but was never mentioned which resolution is actually taken for the analysis; this is mentioned in the abstract only.

Section 2.2

P7/L14: When referring to 'mean concentration', is it temporal mean and if so for which time period? Please describe this step.

P7/L24-25 Here the authors refer to clustering of data between 1989 and 1991. Please describe how was the clustering obtained and why these 3 years and not any other period.

Title for section 3 should only refer to results.

Section 3.1

Please consider to add a table with the average values and STD for each region, this would improve readability for quick check and cross comparison and avoid a long text.

P8/L18 Consider to enumerate the species considered in this section.

P9/L4 Did the authors mean "spatially uniform"?

P9/L16 Please add references to corroborate your assumption.

P9/L8-13 There is a degree of repetition in this paragraph, suggestion that it would fit best at the beginning of the section, before going into detail about each region.

Section 3.2

P10/L16 Discrepancy between the time period described in the title and the introduction of the work.

P10/L19 Why was the year 2000 chosen for normalization?

P10/L20 Typically one would cluster the time series on the basis of similarity, usually comparing magnitude or time variation, but as referred before, nothing was told before how were the stations clustered.

A suggestion for Figure 2 to be moved to the Supplements, and have a figure with average across the region and its variation for both cold and warm. Note that it's hard to read a single plot and across. For figure 3 would be best to put cold and warm seasons together, helping the reader to analyze the results. There is a typo in the legend for $SO_4^{2-}$

Generally, the section lacks which figures the reader should be looking at while reading the results section.

Section 3.3

Generally, the section lacks which figures the reader should be looking at while reading the results section.

Table 3 There is a possible highlighting mistake for concentrations above 1.0 ug m-3, e.g. $NO_3^-$

P17/L10  Please explain why these years were chosen and how the averaging was calculated.

P19/L25-26 The last sentence ("the difference...") is importantly mentioned but the only discussion being written so far. To keep it consistent, it should be moved to the discussion.

P22/L18 How was the normalization of the annual means done? Is it related to the year 2000? Please describe.

Figure 4 does not describe which of the panels is depicted the pollutants.

Figure 5 Please explain why NO3 shows a different pattern that the remaining species.

**Section 4**

P23/L2-3 here is explained why the year 2000 is chosen, please revise the structure so this information is prior to the results.

**Section 5**

P23/L16 Please revise section title.

P23/L19 Why did the authors chose RSO2 instead of 1-RSO4?

Figure 8b has a typo in the legend, and displaying the R would be a good addition

**Section 6**

The paper is already long, it does not need a summary.  PL11-L20 is a section on their own, as there is no real description of the emission in the Eat-US and Canada.

P30/L23 again inconsistency between periods

---

## Author Comment (AC1) · 4 Dec 2019

**Reply to the comments and suggestions of reviewer #1**

*The study presented is very rich in data and analysis. However, some of the information could be summarized making the manuscript more readable, especially in the results section. Every sub-section of the results section has a summary after the description of the results, the summary is good but the text prior to it is overwhelming and unnecessary when having adequate figures/tables supporting the text. The figures and*

*tables should be more carefully thought through, as it takes a lot of time to grasp the information.*

Re: Following the suggestions from the reviewer and Tom Butler (Reviewer 2), we have substantially revised the text to make it more concise and readable. The graphs and tables have been rearranged, eg. Fig. 2 has been moved to Supplemental materials, and a more concise Fig. 2 (as suggested by the reviewer) has been added.

*The reader might have issues to grasp what was done every single step of the way to obtain the results and the analysis, as the methodology is not really clear. This might be a consequence of not having a dedicated section for the methodology. Additionally, it might be challenging the reading when interchanging time periods and not giving a clear context, especially when trends and changes in concentrations are calculated based on different time period, and not really explained why.*

Re: Following the suggestions of the reviewer in the detailed part of the review, we have substantially revised the text to have a clearer context in presenting the results, and explained, for example, why we calculated 10 and 25 years of changes. We also added a dedicated subsection "Statistical analysis and method" to explain how the analysis was carried out.

*More information is needed to understand the levels of concentration behind the clusters, this might be due to emissions, meteorology, terrain, etc. It is important to know if one cluster can be more populated than others in terms of sources, and if that is still the case throughout the study period.*

Re: The fundamental justifications for clustering of 4 regions are: (1) difference in emissions of $SO_2$, $NO_x$ and $NH_3$; (2) the latitudinal gradient of the region, which affects the temperature and the solar radiation; (3) the prevailing atmospheric circulation due to mid-latitude Rossby waves. Regions 1 and 4 can be separated from Regions 2 and 3 because of the low emissions of $SO_2$, $NO_x$ as well as $NH_3$, and also their low/high latitudes; Regions 2 and 3 can be separated from each other due to the large difference

in NH3 emissions. We used ambient concentration of SO2 during the cold season to indicate the difference in SO2 emissions; used NO3 and RNO3 during the cold season to indicate the difference in NH3 emissions.

To support the justification of the clustering, we have added a table in Supplemental materials to show the correlation coefficients of the ambient annual concentration of each site vs. the regional averaged value for each species. For most site and species (except for NO3), the correlation coefficients are larger than 0.95. Also as shown in the time series (Fig. S2), the annual concentrations of sites within each region are highly correlated, which is a demonstration that the clustering of sites is successful.

**Responses to detailed comments:**

*Section 1 P2/L20: the reference to WHO fact sheets should be substituted by papers available describing the impact of air pollution on human health.*

Re: We have substituted the web link by the journal paper references.

*P3/L4-P4/L25: the authors appropriately describe how emissions have been evolving throughout the study period and the legislation to enforce such changes, but there is no single reference to corroborate the numbers. Please add the references grounding your narrative. Additionally, only the US situation was described, there is similar discourse for Canada. Please revise.*

Re: References have been added, and the discussion for Canadian emissions has been added.

*P5/L1-4: the authors describe trend analysis studies for other regions than US, and this might be misleading, even unnecessary. Please consider to remove.*

Re: The mentioned part has been removed, and the text has been revised.

*Section 2.1*

*P6/L25 Please add a reference describing the CASTNET network*

[Figure]

*The authors describe the temporal resolution of the measurements available for each network, but was never mentioned which resolution is actually taken for the analysis; this is mentioned in the abstract only.*

Re: The references for CASTNET have been added. How annual means were derived was added in the new "Statistical analysis and method".

*Section 2.2 P7/L14: When referring to 'mean concentration', is it temporal mean and if so for which time period? Please describe this step.*

Re: We have specified it was "3-years mean concentration of NO3 and SO2 of each site".

*P7/L24-25 Here the authors refer to clustering of data between 1989 and 1991. Please describe how was the clustering obtained and why these 3 years and not any other period. Title for section 3 should only refer to results.*

Re: The method of how to cluster the sub-regions was now clearly specified in the text; We also added an extra table (Table S.3) in Supplemental materials, showing the correlation coefficients of the annual concentration from each site vs. the averaged value of the cluster for each species. The text has been accordingly revised (P9 L18-24).

We have specified that we used the mean concentrations at the beginning of the period because SO2 was highest and it clearly separated regions 1 and 4 from regions 2 and 3.

*Section 3.1 Please consider to add a table with the average values and STD for each region, this would improve readability for quick check and cross comparison and avoid a long text.*

Re: It has been added in Supplemented materials as Table S.1 and S.2.

*P8/L18 Consider to enumerate the species considered in this section.*

Re: It has been revised accordingly (P10 L3-4).

*P9/L4 Did the authors mean "spatially uniform"?*

Re: We specifically mentioned "was spatially uniform".

*P9/L16 Please add references to corroborate your assumption.*

Re: References have been added, and more corroboration has been added. (P10 L24 – P11 L2)

*P9/L8-13 There is a degree of repetition in this paragraph, suggestion that it would fit best at the beginning of the section, before going into detail about each region.*

*Section 3.2 P10/L16 Discrepancy between the time period described in the title and the introduction of the work.*

Re: It was corrected.

*P10/L19 Why was the year 2000 chosen for normalization? P10/L20 Typically one would cluster the time series on the basis of similarity, usually comparing magnitude or time variation, but as referred before, nothing was told before how were the stations clustered.*

Re: The text explaining why year 2000 was chosen for normalization was added (P12 L10-13). In the Section 2.3, we have substantially improved the justification of clustering. Essentially it was based on similarity (correlation coefficients, Table S.3) and magnitude of time variation.

*A suggestion for Figure 2 to be moved to the Supplements, and have a figure with average across the region and its variation for both cold and warm. Note that it's hard to read a single plot and across. For figure 3 would be best to put cold and warm seasons together, helping the reader to analyze the results. There is a typo in the legend for SO42- Generally, the section lacks which figures the reader should be looking at while reading the results section*

Re: Fig. 2 has been removed to the Supplemental materials, and a new Fig. 2 (as suggested) has been added. Fig. 3 has been revised according to the suggestions. We also specified in the revised text the results presented in this section are mainly based on Fig. S3 and Fig. 3

*Section 3.3 Generally, the section lacks which figures the reader should be looking at while reading the results section.*

Re: We specified the results presented in Section 3.3 are summarized from Tables 2 and 3.

*Table 3 There is a possible highlighting mistake for concentrations above 1.0 ug m-3, e.g. NO3-*

Re: The mistake was corrected.

*P17/L10 Please explain why these years were chosen and how the averaging was calculated.*

Re: The explanation has been added, and text has been revised accordingly (P19 L6-L14)

*P19/L25-26 The last sentence ("the difference…") is importantly mentioned but the only discussion being written so far. To keep it consistent, it should be moved to the discussion.*

Re: We have revised the text, and added two more references for the extra reduction of NOx during O3 season. Because of lack of emission data, it is difficult to have a more in-depth discussion.

*P22/L18 How was the normalization of the annual means done? Is it related to the year 2000? Please describe.*

Re: We explicitly mentioned that "The regressed trends are normalized to the regressed value of year 2000 as this is the turning point for the trend of NO3-".

*Figure 4 does not describe which of the panels is depicted the pollutants.*

Re: The mistake was corrected.

*Figure 5 Please explain why NO3 shows a different pattern that the remaining species.*

Re: In the caption of Fig. 5, we explicitly mentioned that "The dot lines link the annual concentrations from 1990 to 2016 for species except NO3- to show the temporal trends". The reason that we didn't use dot-line for NO3 is that NO3 during cold season in region 3 has no trend.

*Section 4 P23/L2-3 here is explained why the year 2000 is chosen, please revise the structure so this information is prior to the results.*

Re: It was revised accordingly.

*Section 5 P23/L16 Please revise section title.*

Re: The title has been revised to "4.1 RSO4 and correlations of RSO4 vs. SO2".

*P23/L19 Why did the authors chose RSO2 instead of 1-RSO4?*

Re: We don't know why "Sickles and Shadwick (2015) chose RSO2. For us, because we are concerned about the fraction of SO2 being oxidized to SO4, so RSO4 is a more direct metrics than RSO2.

*Figure 8b has a typo in the legend, and displaying the R would be a good addition*

Re: The typo has been corrected. R2 has been added in the graphs.

*Section 6 The paper is already long, it does not need a summary. PL11-L20 is a section on their own, as there is no real description of the emission in the Eat-US and Canada.*

Re: We have corrected it accordingly.

*P30/L23 again inconsistency between periods.*

Re: The inconsistency has been corrected.

We sincerely thank the reviewer for his/her exceptionally detailed, very careful, thoughtful, and constructive comments and suggestions. We thank the reviewer for putting time and effects and helping to improve the quality of the manuscript.

Please also note the supplement to this comment:
https://www.atmos-chem-phys-discuss.net/acp-2019-567/acp-2019-567-AC1-supplement.pdf

---

## Author Comment (AC2) · 4 Dec 2019

**Reply to the comments and suggestions of Reviewer #2 (Tom Butler)**

First, we would like to thank Tom Butler (the reviewer) sincerely for his very detailed, thoughtful and constructive review and comments. It helps to improve the quality of the manuscript significantly.

Answers to specific comments:

[Figure]

Comments 1-7: We have revised the manuscript accordingly.

Comment 8: We agree with the reviewer that the Section 3.2 has too much detail on the description of year-to-year variations. We have done significant revisions to remove and skip the minor changes that don't address the overall patterns. We are very appreciate of the reviewer's suggestions.

Comments 9-18: We have revised the manuscript accordingly.

Comment 19: We agree that the original texts are not very clear, and can cause confusion to readers. We have revised the text to indicate that the correlations between SO4 and SO2 can be described by linear regressions for 1990-2010 and 2010-2016 respectively, but the correlation between SO4 and SO2 for the whole study period of 1990-2016 needs to be described by a power-law regression. This helps to explain why in the past some studies suggest that the relationship between SO4 and SO2 be linear, while others suggest that it be a power-law relationship.

Comments 20-29: We have revised the manuscript according to the suggestions.

Comment 30: We tried to combine the 2nd and 3rd graphs into one, but the lines cross over each other, and it is difficult to see each line, so we leave 2nd and 3rd graphs as separate ones.

General comments: We have taken the suggestions of the reviewer, and rounded up some percentage numbers with decimals to integers.

Please also note the supplement to this comment:
https://www.atmos-chem-phys-discuss.net/acp-2019-567/acp-2019-567-AC2-supplement.pdf

---

## Referee Report (RR1)

Some minor comments on Feng et al.    24 January 2020

Throughout the paper, in the text and in many figures (e.g. Figs. 2,3,5,8,9,10,12,13) talk about annual concentrations for the warm and cold seasons.  Technically these would be "seasonal" concentrations rather than "annual".  However, the reader probably understands that. Never-the-less, "seasonal" would be a more accurate term.

Pg 3 Line 17 the Clean Air Act….

Pg 4 Line 22-26   The sharp increase in NOx emissions in the USA is due to a methodology change, not a real drastic change in NOx emissions.  See National Emissions Inventory, https://www.epa.gov/air-emissions-inventories/air-pollutant-emissions-trends-data The change in NOx methodology from 2001 to 2002 is described in the "README" tab on the "Average Annual Emissions" national emissions trends spreadsheet. (I have also discussed this matter with an NEI person.)

Here is a statement in that file:

 **Updates since June 12, 2012**:  Now using NEI 2008 v3 at the Tier 1 level.
 2006 and 2007 were recalculated using interpolation between NEI 2005 v2 and NEI 2008 v3.
 2002 and 2005 MOVES data were used to update 2002-2007.  The change in model
 resulted in noticeable changes in highway emissions from 2001 to 2002 for various pollutants

 NEI does the same thing with NH3 emissions 2000 to 2001, only in that case it is a significant decline in NH3 emissions due to methodology change.

Pg 14 line 3    ….the period was still significant, …..

Pg 17 Line 3    ….or a weak increasing …..

Pg 27 line 9  should this be, "The disparity of the reduction of SO2 and SO4=" ?

Pg 27 lines 15 & 16    "…. but also a larger fraction of SO2….."

Pg 29 lines 3 and 4   correlations are shown on Figure 9, not Figure 10

Pg 29 line 13   "…. sea salts …."

Pg 32 line 22  With the implementation of Title IV of the 1990…..

Pg 34 line 23  "… NH3 in the region was in excess….."

Figure S1 & S2  have much detailed information in them, but the legends are nearly useless unless (as recommended) it is enlarged at least 300%.  It would be helpful if the species of concern is pasted on each map so it can be read at 100%. It would be very useful if the figure legends could be blown up some more. What if someone only has a hard copy of this manuscript? Also the species listing in the

figure caption is not in the same order as they appear as maps. It would be good to have the listing in the caption follow the layout of the maps.

Figs. S3, S4, S5   Again, if the figure legends could be enlarged, it would be very helpful.

Overall, I liked this manuscript. There is a lot of material here, and it is a nice summary and analysis of eastern North America air quality data during a period of generally large emission changes of SO2 and NOx.

---

## Author Response (AR2)

Dear editor,

Here are our responses to the reviewer (Tom Butler)'s minor comments:

1.  We have adopted Tom's suggestion. We have changed the "annual concentrations for the warm and cold seasons" to "seasonal concentrations for the warm and cold seasons" through the text and graphs. To be precise, we also added one sentence in the text to mention that the seasonal concentration was calculated for the warm/cold season for each year:

    *To be precise, the seasonal mean concentrations in this study refer to the mean concentrations calculated for the warm (May – October) and cold (November – April) seasons for each year.*

2.  The increase of NOx emissions in the US in 2002 due to a methodology change has been explicitly mentioned in the text, and the reference was provided:

    *Note that there was a change in NOx measurement methodology from 20001 to 2002, and it caused a sharp increase in the reported NOx emissions in the US from 2001 to 2002 (EPA, 2019).*

3.  All typos have been corrected.

4.  Figure S1 & S2: We took Tom's suggestion, and added a species name to each graph to make the graph readable without enlargement. The order of species in the captions has been rearranged to be consistent with the order of the graphs.

Once again, we sincerely thank Tom and another reviewer for their very detailed and valuable comments and suggestions.

Best regards,

Jian